# Realizing high-power and high-capacity zinc/sodium metal anodes through interfacial chemistry regulation

Zhen Hou [1], Yao Gao[1], Hong Tan[1] & Biao Zhang [1,2,3✉]

Stable plating/stripping of metal electrodes under high power and high capacity remains a great challenge. Tailoring the deposition behavior on the substrate could partly resolve dendrites' formation, but it usually works only under low current densities and limited capacities. Here we turn to regulate the separator's interfacial chemistry through tin coating with decent conductivity and excellent zincophilicity. The former homogenizes the electric field distribution for smooth zinc metal on the substrate, while the latter enables the concurrent zinc deposition on the separator with a face-to-face growth. Consequently, dendrite-free zinc morphologies and superior cycling stability are achieved at simultaneous high current densities and large cycling capacities (1000 h at 5 mA/cm$^2$ for 5 mAh/cm$^2$ and 500 h at 10 mA/cm$^2$ for 10 mAh/cm$^2$). Furthermore, the concept could be readily extended to sodium metal anodes, demonstrating the interfacial chemistry regulation of separator is a promising route to circumvent the metal anode challenges.

[1] Department of Applied Physics, The Hong Kong Polytechnic University, Hong Kong, China. [2] The Hong Kong Polytechnic University Shenzhen Research Institute, Shenzhen, China. [3] Guangdong-Hong Kong-Macao Joint Laboratory for Photonic-Thermal-Electrical Energy Materials and Devices, Research Institute for Smart Energy, The Hong Kong Polytechnic University, Hong Kong, China. ✉email: biao.ap.zhang@polyu.edu.hk

A queous rechargeable batteries are promising alternatives to conventional Li-ion batteries for large-scale energy storage systems[1,2]. The utilization of water solvent offers the advantages of low cost, high safety, and environmental benignity[3,4]. Besides, high ionic conductivities of aqueous media result in superior power densities[5]. Among various aqueous batteries, zinc metal batteries have attracted extensive attention, because Zn anodes are prized for high volumetric capacity (5851 mAh/cm³), abundant resources, and environmental friendliness[6–8]. Although primary Zn metal batteries (e.g., Zn-MnO₂ cells) have been commercialized and used, the practical application in rechargeable batteries is hindered by the formation of Zn dendrites[9,10]. It is well known that during Zn deposition, an uneven electric field distribution inevitably forms at the pores of the separator (Fig. 1a, c)[11,12]. In addition, both $Zn^{2+}$ and the electric field tend to concentrate at the protuberances with high surface energy[13]. Therefore, zinc nucleation and growth prefer to occur at such tips (i.e., "tip effect")[13], resulting in inhomogeneous Zn deposition. The formed Zn protrusions further increase the local electric field intensity around them, leading to the evolution of Zn protuberances into Zn dendrites upon cycling. In particular, high current density and large cycling capacity significantly exacerbate the formation of Zn dendrites, thus dramatically restricting the power and energy density of Zn metal batteries[14]. Specifically, at higher current densities, the $Zn^{2+}$ around the electrode/electrolyte interface are rapidly depleted[15–17]. Subsequently, the dendrites are formed and quickly render a short circuit of cells. Higher cycling capacity would bring about greater volume change, which leads to the pulverization of Zn foil and may make Zn dendrites lose contact with the electrodes, becoming "dead Zn"[13,18]. This would finally give rise to a low reversibility for the Zn deposition/stripping process.

Various strategies have been developed to tackle the above issues, including designing hierarchical electrode structures[19–25], modifying electrolyte formulations[26–30], and optimizing charge/discharge protocols[31,32]. Constructing advanced electrode/electrolyte and separator/electrolyte interfaces are also regarded as an effective approach to stabilize Zn metal anodes through controlling the Zn deposition behavior[33–43]. For example, Kang and colleagues[33] reports that an indium layer coated Zn metal anode could bring about the uniform $Zn^{2+}$ flux and anti-corrosion capability, thus enabling a running lifetime of ~500 h at 1 mA/cm² and 1 mAh/cm². Recently, vertical graphene is introduced into the separator to effectively homogenize electric field distribution and $Zn^{2+}$ flux[41], which suppresses the Zn dendrites growth and realizes improved cycle life of ~80 h (5 mA/cm², 5 mAh/cm²) and ~600 h (10 mA/cm², 1 mAh/cm²).

Despite these exciting progress, achieving long-term cyclic stability for Zn plating/stripping remains a great challenge, especially at simultaneous high current density (>5 mA/cm²) and large cycling capacity (>5 mAh/cm²)[14]. At present, most of the reported works stabilize Zn metal anodes through suppressing the Zn dendrite growth. However, the formation of Zn dendrites is inherently unavoidable, notably at rigorous testing conditions, as such a process is thermodynamically and kinetically favorable[15,44]. Therefore, it is highly desirable to develop a unique strategy that could not only restrain the Zn dendrites initiation but also eliminate inevitably formed Zn dendrites.

In this work, we construct metallic Sn-coated separator via magnetron sputtering to stabilize Zn metal anodes. The highly conductive Sn coating with excellent zincophilicity could help to homogenize $Zn^{2+}$ flux (as simulated by the finite element method) and meanwhile manipulate the growth direction of Zn metal. Thanks to these synergetic effects, dramatically improved cycle life of 1000 h (5 mA/cm², 5 mAh/cm²) and 500 h (10 mA/cm², 10 mAh/cm²) are realized on Zn/Zn symmetric cells. Furthermore, we demonstrate that the approach could be readily extended to Na/K metal anodes for enabling safe and high-performance Na/K metal batteries.

## Results

**Mechanism of controllable dendrites growth.** We propose the use of separators modified with conductive and zincophilic coatings to build smooth Zn deposition on both the anode and separator, which helps to stabilize Zn metal anodes at high current densities and large cycling capacities. A nonuniform electric field distribution commonly exists due to the concentrated electric field on the pores of pristine separator[11]. The introduction of conductive layer on the separator is expected to homogenize the electric field between the separator and the anode due to the equipotential surface of the conductor[45,46]. The excellent zinc affinity of the coating would be beneficial to facilitating the uniform transport of $Zn^{2+}$ towards the anode under the homogeneous electric field[47,48]. Effect of the conductive coating on

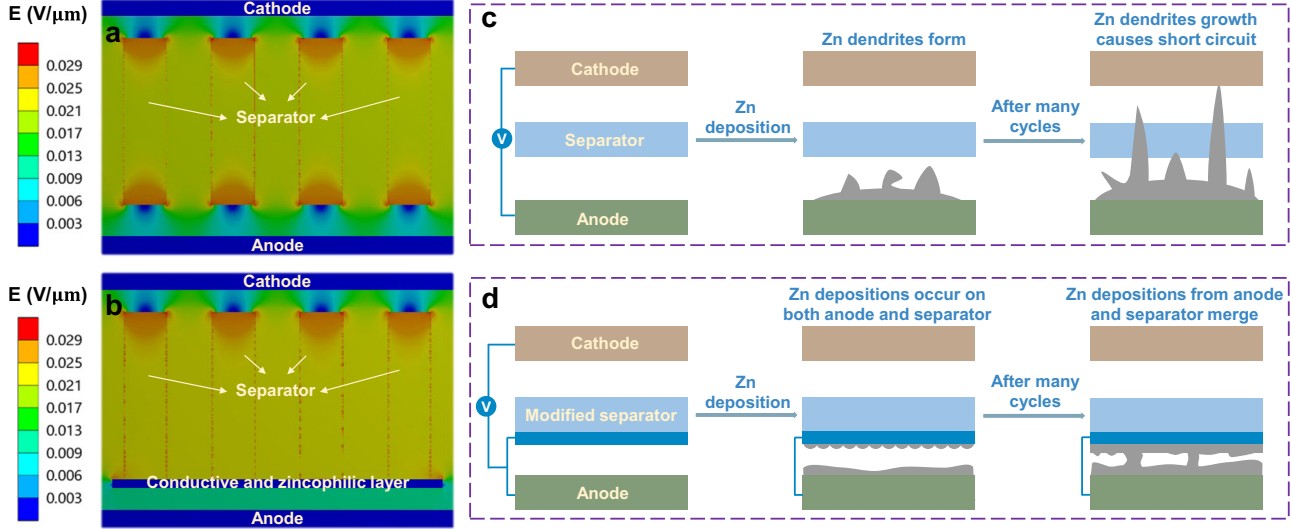

**Fig. 1 Theoretical calculation and protection mechanism of modified separator. a** Electric field distribution with the pristine separator. **b** Electric field distribution in the non-contact region of the modified separator and the anode. **c** Schematic illustration of Zn deposition with the pristine separator. **d** Schematic illustration of Zn deposition in the contact region of the modified separator and the anode (see Supplementary Note 1 for details).

electric field distribution is simulated by the finite element method (Supplementary Figs. 1–3 and Supplementary Note 1). As shown in Fig. 1a and Supplementary Movie 1, the electric field intensity on the pores of pristine separator is obviously higher than that of the adjacent cellulose skeletons, which would give rise to severely nonuniform Zn deposition on the anode. Such preferential Zn deposition further evolves into Zn dendrites through self-amplification mechanism, eventually causing the internal short circuit of cells (Fig. 1c). On the contrary, a uniform electric field is observed after introducing the conductive layer on separator due to its equipotential surface (Fig. 1b and Supplementary Movie 2). Notably, such enhancement is applied for the region where the Sn coating is not in contact with the anode. The zincophilic nature of the coating would help to generate the homogenous $Zn^{2+}$ flux under this uniform electric field, in turn enabling a smooth Zn deposition on the anode. Meanwhile, within the region where Sn coating is in contact with the anode, a highly zincophilic coating layer will trigger the concurrent Zn deposition on the separator (Fig. 1d)[49,50]. Upon cycling, the Zn depositions from the anode and the separator will meet and merge, enabling a compact morphology. Furthermore, the direction of Zn growth changes from perpendicular to parallel, to the separator, which prevents it from piercing through the separator.

**Screening and fabrication of appropriate coatings on separator.** Based on the above-proposed mechanism, a superior coating on separator should meet the following requirements: (1) be water insoluble and have a higher redox potential than Zn metal to avoid electrochemical oxidation; (2) possess zincophilic nature for attracting $Zn^{2+}$ to coating/electrolyte interface; and (3) have a decent electric conductivity to yield an equipotential surface and provide electrons for reducing $Zn^{2+}$ into Zn. Thus, inherently conductive metals with an applicable redox potential and favorable zincophilicity would be promising coating candidates. To screen the ideal metal elements, we compare the zincophilicity of various metals with eligible redox potentials, including Sn, Ag, Bi, and Sb. Metal slurries are cast on Ti foils to fabricate corresponding metal-modified Ti foils (denoted as metal-Ti), to investigate their zincophilicity. The value of nucleation overpotential ($\eta$) is used to evaluate the zincophilicity of these species. Here, $\eta$ is calculated according to the potential difference between the voltage tip and subsequent stable voltage[51]. As shown in Fig. 2a, Supplementary Fig. 4, and Supplementary Note 2, $\eta$ of the bare Ti current collector is 44 mV. The Sn-Ti current collector presents the lowest $\eta$ of 8 mV among these metals, followed by Ag-Ti (16 mV), Bi-Ti (19 mV), and Sb-Ti (49 mV). Therefore, the Sn element will be an optimal candidate to prepare the high-quality coating on the separator. To figure out the reason of the excellent affinity between Sn and Zn, the electrochemical behavior of Sn-Ti/Zn cell is investigated. First, the cyclic voltammetry (CV) curve of Sn-Ti/Zn cell between 0.01 and 0.6 V is collected to show whether there are any alloy reactions between them. A cathodic peak at ~0.28 V and two anodic peaks at 0.38 and 0.47 V are observed (Fig. 2b and Supplementary Fig. 5), which should be assigned to the alloying and dealloying between Sn and Zn. The discharge curve of the Sn-Ti/Zn cell is collected at 0.08 mA/cm² (the cut-off voltage is 0.01 V) and it could deliver a discharge capacity of 21 mAh/g (based on the mass of Sn) (Fig. 2c), which further confirms the alloy reaction and explains the superb affinity between Zn and Sn[52]. The alloy reaction results in the X-ray diffraction (XRD) peak shift of pristine Sn to lower angle after discharge (Supplementary Fig. 6). Furthermore, the X-ray photoelectron spectroscopy (XPS) spectra of pristine and discharged Sn also reveal alloy reaction (Fig. 2d). The formation of Zn-Sn

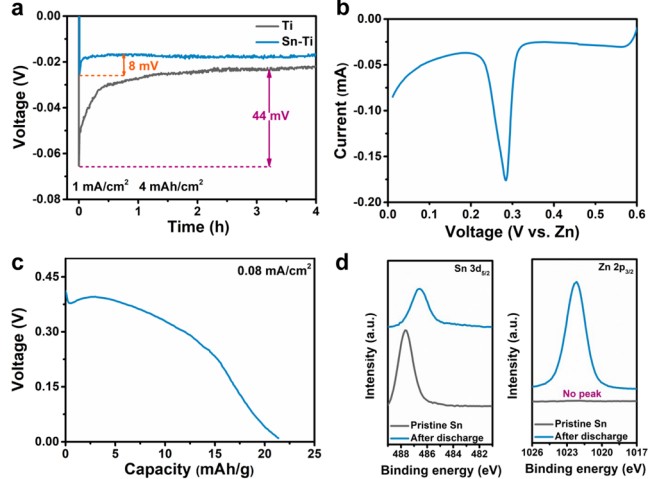

**Fig. 2 The alloy reaction between Sn and Zn. a** The nucleation overpotential of Zn on Ti and Sn-Ti current collectors; **b** CV of Sn-Ti/Zn cell at a scan rate of 0.2 mV/s; **c** the galvanostatic discharge curve of Sn-Ti/Zn cell at 0.08 mA/cm²; and **d** XPS spectra of Sn 3d and Zn 2p for pristine and discharged Sn.

alloy alters the electronic states of Sn, bringing about a downshift of the Sn $3d_{5/2}$ peak from 487.6 eV for the pristine one to 486.6 eV after discharge[53,54]. Meanwhile, a new Zn $2p_{5/2}$ peak at 1022.2 eV is detected in the discharged sample, confirming the formation of Zn-Sn alloy.

Magnetron sputtering is easy to handle and is reproducible, allowing a precise thickness control (Supplementary Fig. 7 and Supplementary Note 3) of the film at the nanoscale level[55]. These characteristics give it unparalleled advantages to prepare Sn-coated separator. We focus on the modified separator with a sputtering time of 1 min, because it has the best performance in stabilizing Zn metal anodes. As seen from optical images (Fig. 3a, b), a gray Sn coating is constructed on the white pristine separator after sputtering. The average mass loading of Sn coating is only ~0.06 mg/cm², which is <3% of the pristine separator (~2.14 mg/cm²), having a negligible impact on the energy density of the battery. The scanning electron microscopy (SEM) images show that the separator is uniformly covered by Sn particles with an average diameter of ~60 nm. The corresponding energy dispersive spectroscopy mapping further proves the Sn elements are evenly distributed on the separator (Supplementary Fig. 8). The peaks of Sn-coated separator in XRD are assigned to tetragonal Sn (JCPDS#65-2631) (Fig. 3c). The stability of such Sn layer in electrolyte is assessed through comparing the XPS and Raman spectra of pristine Sn-coated separator with that after 20 cycles. It is observed that the Sn metal on the separator is highly resistant to oxidation in the aqueous electrolyte (see Supplementary Fig. 9 for details). The $\eta$ value of Ti/Zn cell with pristine and Sn-coated separator are compared, to demonstrate whether the Zn deposition could simultaneously take place on the modified separator. As shown in Fig. 3d, the Ti/Zn cell with Sn-coated separator has an $\eta$ of 28 mV, which is much lower than that using pristine one (44 mV). Such a small $\eta$ reflects that the Zn deposition should be initiated on the zincophilic Sn-coated separator instead of the Ti current collector.

**Morphologies of Zn deposition on the Sn-coated separator.** Above results indicate that Zn would deposit on the Sn-coated separator. To confirm this conjecture, SEM images of Ti current collector and Sn-coated separator retrieved from cycled Ti/Zn cells are collected. Under a deposition capacity of 1 mAh/cm²,

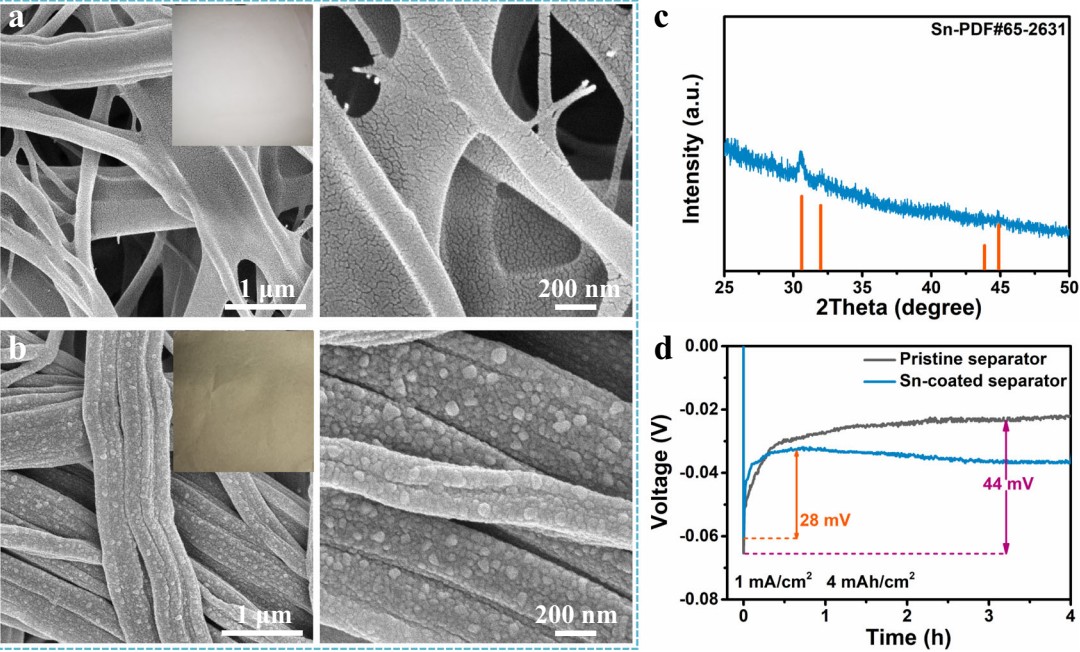

**Fig. 3 Characterization of Sn-coated separator.** The SEM images of **a** pristine separator and **b** Sn-coated separator with the corresponding optical photos in the insets. **c** The XRD spectrum of Sn-coated separator. **d** The nucleation overpotential of Ti/Zn cell with pristine separator and Sn-coated separator.

loose structures with uneven Zn chaotic clusters are observed on Ti current collector using the pristine separator (Fig. 4a). With an increasing deposition capacity of $4 \, mAh/cm^2$, Zn maintains similar dreadful morphology with sharp tips presented (Fig. 4b), which might pierce the separator upon cycling. Such notorious structures may result from the inhomogeneous nucleation and deposition of Zn due to the uneven electric field distribution, which inevitably forms at the pores of the separator (Fig. 1a, c). The uneven deposition of Zn further evolves into Zn dendrites through self-amplification mechanism. By contrast, the Ti current collector in the cell using Sn-coated separator present a much improved Zn deposition behavior. Specifically, dense and uniform Zn coatings are obtained under the capacity of both 1 and $4 \, mAh/cm^2$ (Fig. 4c, d). The dendrite-free morphologies should be attributed to the homogenous electric field enabled by the Sn coating on the separator, which agrees well with the simulation results of finite element method (Fig. 1b). The corresponding SEM images of Sn-coated separator verify that the Zn deposition could be realized on the separator. A smooth layer of Zn is observed on the separator under a deposition capacity of $1 \, mAh/ cm^2$ (Fig. 4e and Supplementary Fig. 10). Moreover, the Zn deposition remains uniform at a higher deposition capacity ($4 \, mAh/cm^2$) (Fig. 4f). Therefore, the Sn coating enables the smooth Zn growth on both the separator and current collector, which will merge at later stages to suppress the dendrite growth.

**Electrochemical performances of Sn-coated separator for Zn metal batteries.** To elucidate the effect of Sn-coated separator, the coulombic efficiency (CE) of Ti/Zn and cycling performances of Zn/Zn symmetric cells using pristine and Sn-coated separators are evaluated. CE is a critical parameter to evaluate the reversibility of Zn deposition/stripping, which is calculated based on the capacity ratio of stripping to plating. The cells using Sn-coated separator show high and stable CE values compared with that using pristine separator (Supplementary Fig. 11 and Supplementary Note 4). Moreover, a rapid short circuit is observed on the cell using the pristine separator, indicating rampant Zn dendrite growth during the plating/stripping process. Therefore,

Sn-coated separator allows for better Zn reversibility and suppresses the growth of Zn dendrites. Turning to cycling stability, as shown in Supplementary Fig. 12 and Fig. 5a, Zn/Zn cell with pristine separator suffers from sudden voltage drop after ~900 h ($1 \, mA/cm^2$ and $1 \, mAh/cm^2$) and ~250 h ($2 \, mA/cm^2$ and $2 \, mAh/ cm^2$), which is ascribed to a short circuit caused by the growth of Zn dendrites. The more rapid cell failure is observed at higher current densities and cycling capacities (Fig. 5b, c). Concretely, the cells experience short circuits at ~170 h ($5 \, mA/cm^2$ and $5 \, mAh/cm^2$) and ~80 h ($10 \, mA/cm^2$ and $10 \, mAh/cm^2$). This is due to the rampant dendrites growth induced by the local electric field intensity and depleted $Zn^{2+}$ concentration at the electrode/ electrolyte interface. In contrast, stable cycle life up to 4500 h and 3800 h with a stable overpotential is respectively realized in the cell with Sn-coated separator at the condition of $1 \, mA/cm^2$ for $1 \, mAh/cm^2$ and $2 \, mA/cm^2$ for $2 \, mAh/cm^2$. Remarkably, Sn-coated separator enables steady cycling for 1000 h at $5 \, mA/cm^2$ for $5 \, mAh/cm^2$. An exceptional Zn plating/stripping life of 500 h could be achieved at a higher current density of $10 \, mA/cm^2$ and cycling capacity of $10 \, mAh/cm^2$, indicating its excellent potential for practical use.

The cycle life of our work and previous reports is summarized in Supplementary Table 1. It is observed that symmetric batteries using Sn-coated separator have absolutely leading cycling stability at multiple current densities and cycling capacities. To better compare the cycling performance of symmetrical cells at different test conditions, the cumulative capacity[56] (current density × cycle life) vs. the per-cycle areal capacity is plotted in Fig. 5d (based on the data summarized in Supplementary Table 1). Figure 5d contains the three key parameters of cycle life, current density, and cycling capacity (per-cycle areal capacity), which can provide a comprehensive picture of the electrochemical performance of symmetric batteries. This work delivers a high cumulative capacity of $7600 \, mAh/cm^2$ ($2 \, mA/cm^2$) at a cycling capacity of $2 \, mAh/cm^2$, surpassing the value in most previous works. Moreover, cumulative capacity of more than $5000 \, mAh/cm^2$ at $5 \, mA/cm^2$ and $10 \, mA/cm^2$ is respectively realized at high cycling capacities of $5 \, mAh/cm^2$ and $10 \, mAh/cm^2$. Notably, such high cumulative capacities cannot be achieved at simultaneously high

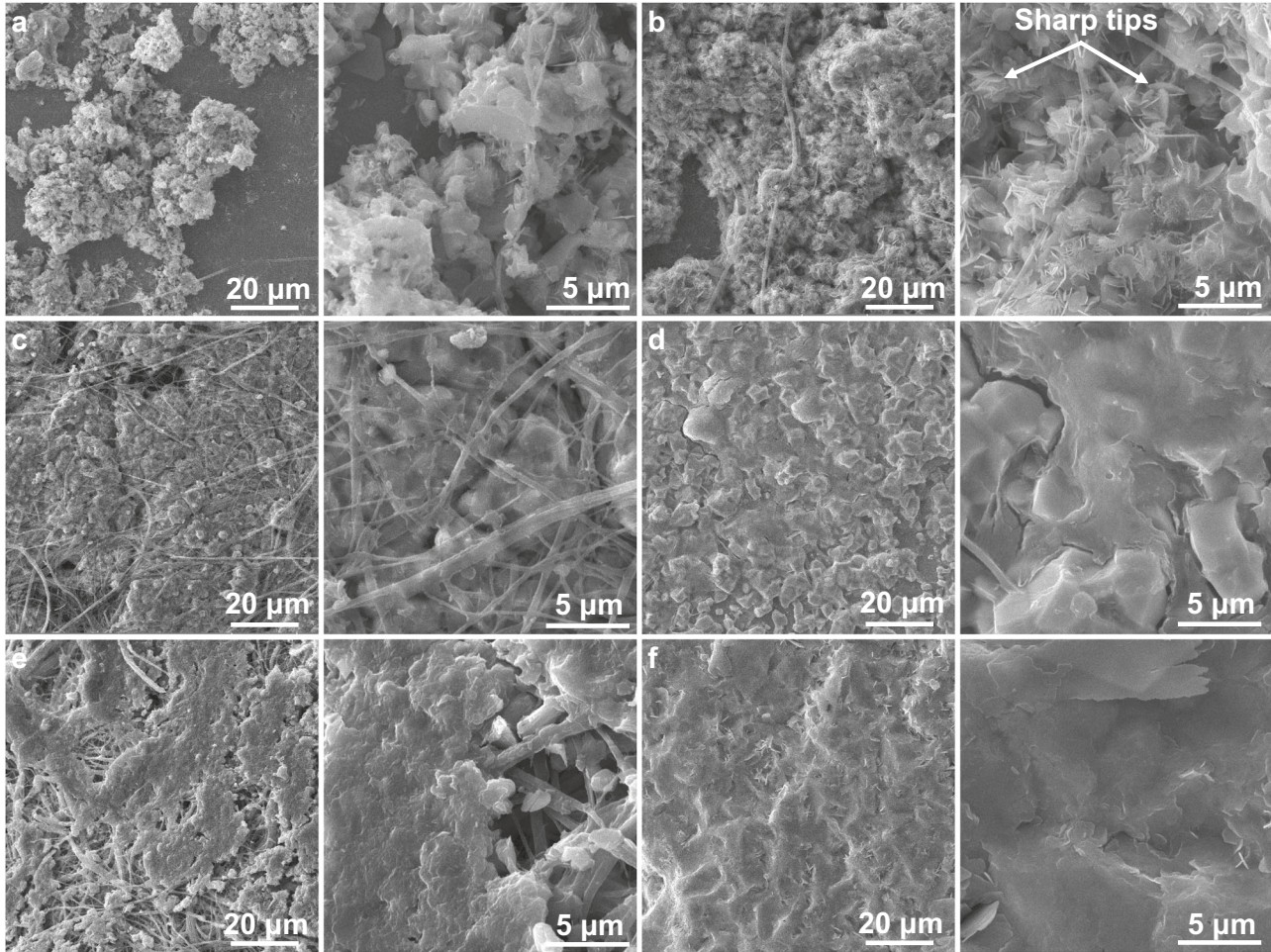

**Fig. 4 SEM images of the electrodes after Zn deposition at 1 mA/cm².** Zn deposition on Ti current collector using pristine separator with a cycling capacity of **a** 1 mAh/cm² and **b** 4 mAh/cm²; Zn deposition on Ti current collector using Sn-coated separator with a cycling capacity of **c** 1 mAh/cm² and **d** 4 mAh/cm²; Zn deposition on Sn-coated separator with a cycling capacity of **e** 1 mAh/cm² and **f** 4 mAh/cm².

current density and large per-cycle areal capacity in previously reported approaches. These results clearly demonstrate the great superiority of our technique at a wide range of current densities and cycling capacities.

The mechanism behind improved cycling stability is further explored by the electrochemical impedance spectroscopy (EIS) and SEM images after cycles. The EIS data are simulated by equivalent circuits (Supplementary Fig. 14). An additional parallel resistor–capacitor circuit is incorporated for Sn-coated separator due to the presence of an extra depressed semicircle that is related to the Sn coating/electrolyte interface. The fitting resistance results are shown in Supplementary Table 2. The charge transfer resistance at the Zn/electrolyte interface ($R_{ct}$) of Zn/Zn cells with pristine separator is 42.14 Ω after 10 cycles and then increases to 73.69 Ω after 20 cycles. In contrast, Zn/Zn cells with Sn-coated separator present stable and much lower resistances. The resistance at the Sn coating/electrolyte interface ($R_{sf}$) and $R_{ct}$ only slightly rise from 5.81 Ω and 2.26 Ω in the 10 cycles to 6.61 Ω and 2.81 Ω after the 20 cycles, respectively. The low and stable interfacial resistances indicate the enhanced deposition/stripping kinetics and interfacial stability. The SEM images after cycling provide further concrete evidence (Supplementary Fig. 15). An uneven surface with many protrusions is observed on Zn anode using pristine separator after one cycle due to inhomogeneous Zn deposition. Moreover, the cycled Zn evolves into a looser and rougher structure after 20 cycles. Benefiting from the uniform

Zn²⁺ flux by Sn coating, Zn anode presents a much smooth and uniform surface after 1 cycle and 20 cycles. Turn to the SEM images of cycled Sn-coated separator, it is partially covered by the flat Zn after one cycle. The dense Zn metal is observed on Sn-coated separator after 20 cycles. The Zn depositions growing from anode and separator are supposed to meet and merge upon cycling, which gives rise to the compact Zn metal layer and changes the Zn growth direction. This is confirmed by dendrite-free Zn morphologies on both Zn anode and Sn-coated separator even after 200 cycles (Supplementary Fig. 16). These results indicate Sn-coated separator could homogenize Zn²⁺ flux and merge Zn deposition from the substrate and separator, which leads to the dendrites-free morphologies and reinforced inter-facial stability, realizing the safe operation of the cells even at rather rigorous testing conditions. It is noteworthy that the superior performance is realized with neither Sn-coated Zn foils nor other metal-coated separator (Supplementary Figs. 17–19 and Supplementary Note 5). It suggests a synergistic effect between the Sn-induced highly zincophilicity and the face-to-face growth in suppressing dendrite growth. Lastly, we explore the effect of Sn coating thickness for stabilizing Zn metal anode by varying the sputtering time (Supplementary Figs. 20–22). The results demonstrate that precise tuning of the Sn coating thickness at the nanoscale level, where magnetron sputtering shows great advantages over classic doctor-blade coating (Supplementary Figs. 23 and 24, and Supplementary Note 6), is essential to

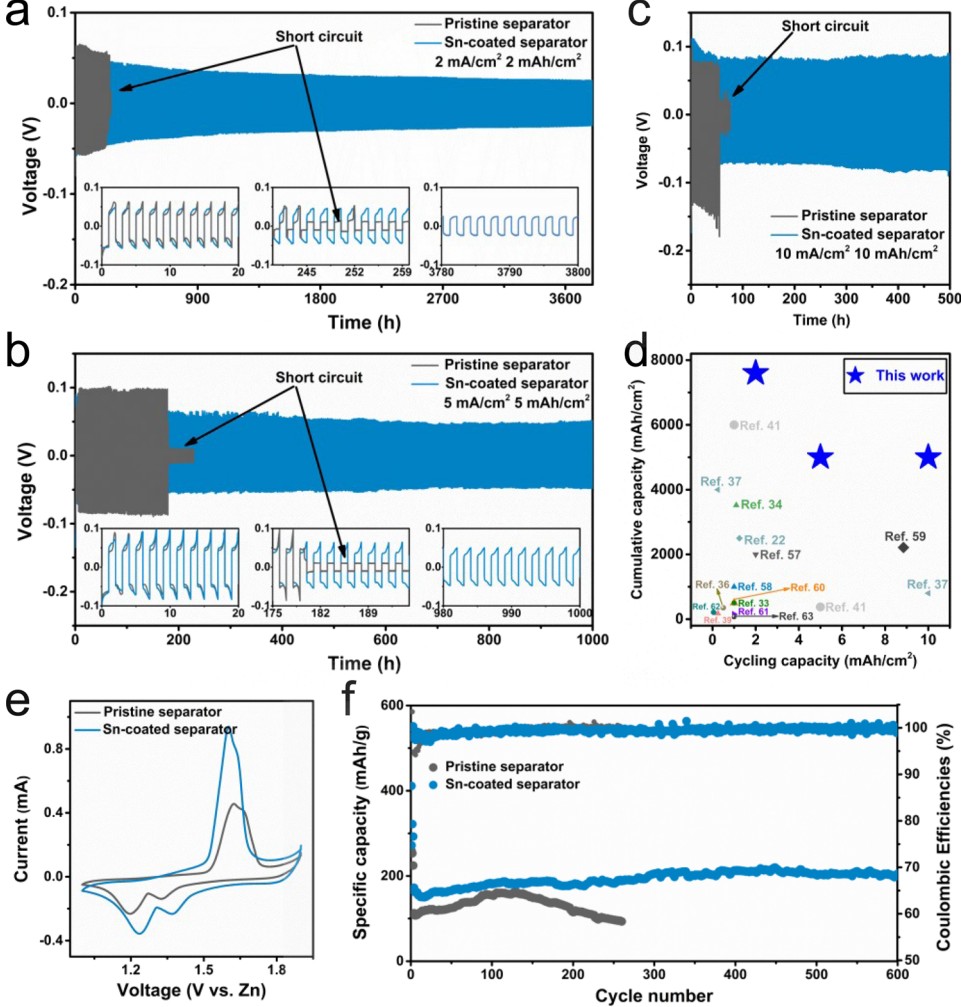

**Fig. 5 The electrochemical performances of Zn metal batteries.** The cycling performances of Zn/Zn cells using pristine separator and Sn-coated separator tested at **a** 2 mA/cm² and 2 mAh/cm², **b** 5 mA/cm² and 5 mAh/cm², and **c** 10 mA/cm² and 10 mAh/cm². **d** Comparison of cycling performance (cumulated capacity vs. cycling capacity) in this work and previously reported works (vertical graphene separator[41]; polyacrylonitrile separator[39]; reduced graphene oxide coated Zn[65]; carbon-coated Zn[66]; PVDF/TiO₂ layer[67]; ZnO coating[22]; polyamide coating[37]; Kaolin layer[34]; Al₂O₃ coating[68]; TiO₂ coating[69]; indium layer[33]; MOF layer[36]; CaCO₃ coating[70]; and polyacrylamide separator[71]). The electrochemical performances of Zn||MnO₂ batteries using pristine separator and Sn-coated separator: **e** CV at a scan rate of 0.1 mV/s (second cycle); **f** cycling performances at 0.3 A/g, with Zn foil as the anode.

achieving the superior performance, especially under stringent test conditions.

To assess the practical application of Sn-coated separator, the full cell paired with a MnO₂ cathode (Supplementary Fig. 25) is assembled. The CV curves of full cells using pristine separator and Sn-coated separator are compared in Fig. 5e. They present the same Mn-ion redox peaks, which agrees well with the previous works[57]. The full cell with Sn-coated separator shows lower oxidation potential, higher reduction potential, and peak current than that with pristine separator, suggesting improved reaction kinetics for Sn-coated separator[34,58,59]. Supportive evidence could be found at the rate performances of full cells (Supplementary Fig. 26). At a high current density of 0.75 A/g (equivalent to 2C), a full cell with the Sn-coated separator can provide a discharge capacity equivalent to approximately 200% of that with the pristine separator (107 mAh/g vs. 53 mAh/g).

The long-term cycling stability of full cells using Zn foil as an anode is evaluated at 0.3 A/g. Benefitting from effective Sn-coated separator, the full cell presents a stable cycle life with a discharge capacity of ~200 mAh/g after 600 cycles. Turning to the one with pristine separator, the discharge capacity is ~159 mAh/g after 130 cycles and then gradually drops to ~94 mAh/g after 260 cycles.

More critically, using a pre-set amount of Zn as the anode, the cycling performance of the full cell is further evaluated at the specific negative-to-positive electrode capacity (N : P) ratios of 10 : 1. The discharge capacity of the full cell with the pristine separator deteriorates rapidly after about 60 cycles, with a capacity of merely ~63 mAh/g after 80 cycles (Supplementary Fig. 27). On the contrary, the highly improved cycle stability is achieved over 180 cycles (discharge capacity of ~145 mAh/g) on the full cell with the Sn-coated separator. Furthermore, we use the commercially available cathode (activated carbon (AC)) to elucidate the benefits of Sn-coated separator. The performance of Zn/AC full cells using Sn-coated separator is tested at 12 A/g. Surprisingly, such cell can stably deliver a discharge capacity of 54 mAh/g for more than 20,000 cycles (Supplementary Fig. 28b). These observations are in line with the improved lifetime of Zn/Zn cell using Sn-coated separator and confirm the benefits in practical electrochemical systems. The performance of our work and other research is summarized in Supplementary Tables 3 and 4. It is found that the stability of our work is competitive with those previous reports.

We note that the Sn element is also highly sodiophilic[60,61]. The strategy of interfacial chemistry regulation may also be applicable

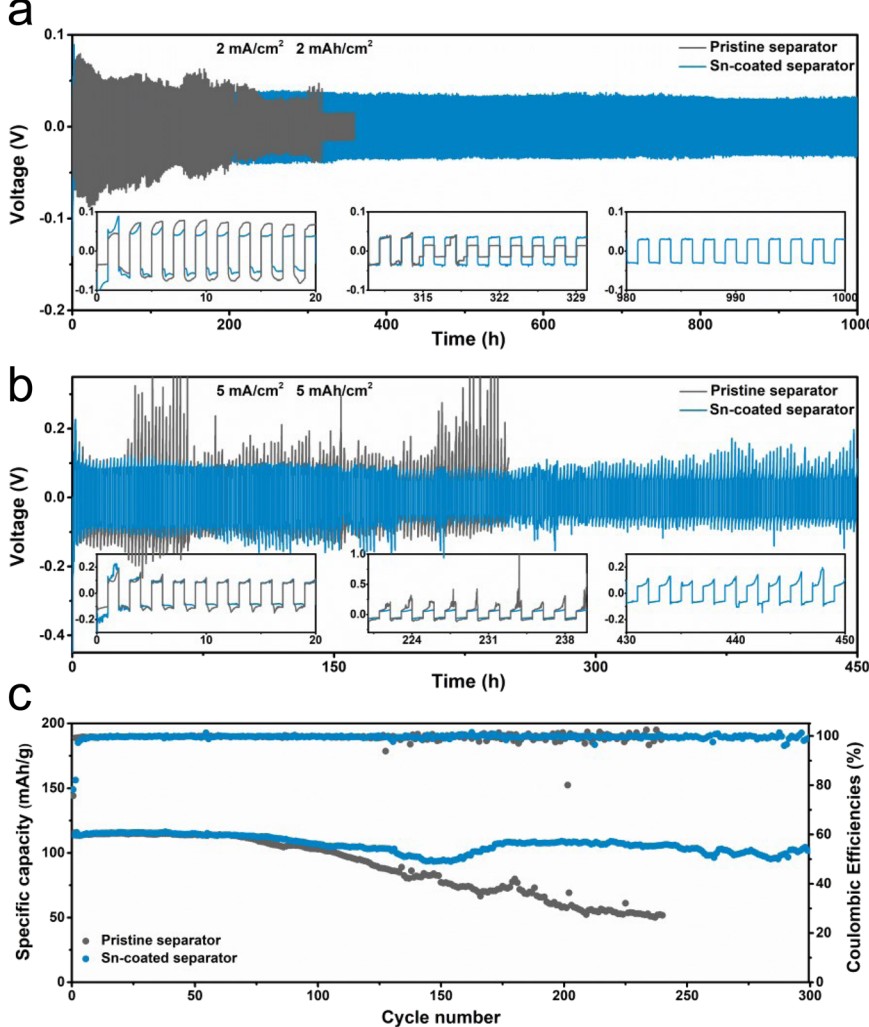

**Fig. 6 The electrochemical performances of Na metal batteries.** The cycling performances of Na/Na cells using pristine separator and Sn-coated separator are tested at **a** 2 mA/cm² and 2 mAh/cm², **b** 5 mA/cm² and 5 mAh/cm². **c** The cycling performances of Na||NVPF batteries using pristine separator and Sn-coated separator at 0.1 A/g, with N : P ratios of 5 : 1.

to Na metal anodes, which suffers from similar safety hazards related to Na dendrites growth[62]. A Na/Na symmetrical cell is fabricated using the Sn-coated separator. After cycling, the Na deposition on Sn-coated separator could be clearly visualized in the optical image (Supplementary Fig. 29). It would trigger the face-to-face growth of Na metal on the substrate and separator, which is beneficial to preventing dendrite growth and improving the cyclic stability, as demonstrated earlier in the Zn metal case. The electrochemical performance adopting Sn-coated separator is evaluated using Na/Na symmetric cells. As shown in Supplementary Fig. 30, a prolonged lifetime of over 1200 h is also realized on the Na/Na symmetry cell using the Sn-coated separator compared to that using the pristine separator (880 h). At a higher current density of 2 mA/cm², the cell with pristine separator fails after ~320 h cycling at a capacity of 2 mAh/cm², whereas a smooth and stable voltage profile for 1000 h is observed in the cell with Sn-coated separator at the same test condition (Fig. 6a). Noticeably, the cell with Sn-coated separator could stably operate for 450 h under a higher current density (5 mA/cm²) and cycling capacity (5 mAh/cm²) with only a slightly increased overpotential (Fig. 6b), which should be attributed to the regulated Na⁺ flux and the capability of merging Na deposition from the anode. In contrast, a dramatically fluctuant voltage profile is displayed in the cell with pristine separator,

owing to the unstable electrode/electrolyte interface and the growth of dendrites. It finally ends up at ~230 h with a huge overpotential up to 1 V. We further apply this strategy to potassium (K) metal anodes and realize a lifetime of 450 h at a large current density (3 mA/cm²) and cycling capacity (3 mAh/cm²), as seen from Supplementary Fig. 31. The benefits of adopting the Sn-coated separator are confirmed in the $Na_3V_2(PO_4)_2F_3$ (NVPF)/Na full cells with a N : P capacity ratio of 5 : 1 (Fig. 6c). The Cu current collectors deposited with a certain amount of Na are employed as anodes. Full cells with Sn-coated separator deliver much improved capacity retention than that using a pristine separator. These results demonstrate the feasibility of Sn-coated separator for Na/K metal batteries.

## Discussion

The dendritic growth of Zn is inherently unavoidable, as the process is thermodynamically and kinetically favorable. In view of that, we propose an advanced separator modified by a conductive and zincophilic coating layer, which not only retards the growth of Zn dendrites but also eliminates the inevitably formed Zn dendrites. After screening various candidate elements, Sn is selected because of its electrochemical stability, electrical conductivity, and excellent zincophilicity. As revealed by the

theoretical simulation, the conductive Sn layer brings about uniform electric field distribution, enabling smooth Zn deposition on the anode and delaying the formation of Zn dendrites. In addition, the electrical conductivity and zincophilicity of Sn coating triggers Zn deposition on the separator. It merges the Zn grown from the anode during cycling, eliminating the inevitably formed Zn dendrites and avoiding short circuits. Consequently, dendrites-free Zn morphologies and highly prolonged cycling performances (1000 h at 5 mA/cm$^2$ for 5 mAh/cm$^2$ and 500 h at 10 mA/cm$^2$ for 10 mAh/cm$^2$) are realized under rigorous testing conditions, allowing the operation of MnO$_2$/Zn full cell with limited excess of Zn metal. Furthermore, this strategy could also be applied to Na/K metal batteries owing to the sodiumphilic/potassiumphilic nature of Sn. This work provides fresh insights for constructing safe metal batteries by modifying the separator/electrolyte interface.

## Methods

**Preparation of Sn-coated separator.** The Sn layer was coated onto a commercially available cellulose separator (thickness of 30 μm) using direct current magnetron sputtering system. The distance between the Sn target and cellulose separator was 10 cm. The sputtering time was set as 0.5, 1, 2, and 5 min at the power of 50 W. The average loading mass of Sn layer was ~0.06 mg/cm$^2$ for sputtering time of 1 min. Similarly, Ag coating was constructed on cellulose separator after sputtering 1 min. The Sn-coated separator was also prepared using the doctor blading technique. The slurry was prepared by uniformly mixing 90 wt% Sn nanopowders (<150 nm particle size) and 10 wt% polyvinylidene difluoride (PVDF) in N-methyl-2-pyrrolidone solvent. The homogeneously mixed slurry was then coated onto the cellulose separator using the doctor-blade method with the minimum thickness grade for the scraper (25 μm) to make the Sn-coated separator. For convenience, the sample is marked as DSn-coated separator.

**Synthesis of metal-modified Ti foils, MnO$_2$, and NVPF.** The slurries were prepared by uniformly mixing 80 wt% metal powder (metals including Sn, Bi, Ag, Sb), 10 wt% Super P, and 10 wt% PVDF in N-methyl-2-pyrrolidone solvent. The metals modified Ti foils (denoted as metal-Ti) were fabricated by casting the corresponding slurries on Ti foil using the doctor-blade method. After drying at 60 °C for 12 h, metal-Ti foils were then punched into circular disks (diameter of 12 mm) prior to use. Similarly, Super P-modified Ti foil (denoted as SP-Ti) was prepared by same procedure, except that the slurry was fabricated by uniformly mixing 90 wt% Super P and 10 wt% PVDF. MnO$_2$ was prepared using the hydrothermal method according to the previous report[57]. Specifically, 2.5 mL of 1.0 M H$_2$SO$_4$ and 158 mg of KMnO$_4$ were mixed with deionized water (30 mL) under magnetic stirring until dissolved. Subsequently, 95 mg of zinc powders were added to the solution and stirred magnetically at 60 °C for 6 h. The product was then collected by filtration and washed with deionized water until the pH value was higher than 6. Finally, the product was dried at 80 °C for 12 h. NVPF was fabricated by a two-step solid-state reaction[63].

**Characterizations.** The coin cells were disassembled to obtain the cycled electrodes. These electrodes were washed by deionized water to remove residual electrolytes before characterization. The SEM images were collected by Tescan VEGA3. The XRD and XPS were recorded by X-ray diffractometer (Rigaku SmartLab) with Cu Kα radiation and X-ray photoelectron spectrometer (Nexsa) with Al Kα X-ray line, respectively. The thickness of the sputtering Sn layer on separator and Raman measurements were respectively carried out on atomic force microscope (Bruker) and Witec-Confocal Raman system (UHTS 600 SMFC VIS) with a laser wavelength of 532 nm.

**Electrochemical measurements.** CR2032 coin-type cells were assembled with 2 M ZnSO$_4$ as an electrolyte in the ambient atmosphere. Zn foils were employed as both the counter and reference electrodes. Ti foil, metal-Ti foil, and Zn foil were adopted as working electrodes to make Ti/Zn, metal-Ti/Zn, and Zn/Zn cells, respectively. The pristine and Sn-coated separator were adopted for comparison. The cycling performances of Zn/Zn cells were performed at various current densities and cycling capacities. For Na and K batteries, 1 M NaPF$_6$ in diethylene glycol dimethyl ether (DGME), and 1 M potassium bis(fluorosulfonyl)imide in DGME were respectively used as the electrolyte. The CV and EIS measurements were obtained using the BioLogic electrochemical workstation (versatile potentiostat). EIS was performed from 10$^5$ and 10$^{-1}$ Hz with a potential amplitude of 5 mV. A slurry consisting of 70 wt% MnO$_2$, 20 wt% Super P, and 10 wt% PVDF was coated on stainless steel to prepare MnO$_2$ cathode. Zn foil and Ti foil with a specific amount of Zn were paired with MnO$_2$ cathode for full cell evaluation. The electrolyte was 2 M ZnSO$_4$ + 0.2 M MnSO$_4$, where MnSO$_4$ helps to suppress Mn$^{2+}$ dissolution[64]. It is found that the MnSO$_4$ additive has a negligible influence on Zn deposition/

stripping behavior (Supplementary Fig. 32 and Supplementary Note 7). These full cells were tested between 0.8 and 1.9 V.

**Theoretical computations.** Finite element method conducted by Ansys was adapted to simulate the electric field distribution with the pristine separator and the conductive/zincophilic separator (Sn-coated separator). The pristine separator was modeled as a sieve plate with a thickness of 3.8 μm, which was composed of rectangular channels with an aperture of 1.0 μm and a hole spacing of 1.0 μm. The potential difference between cathode and anode was set as 0.1 V. The electrical conductivity of anode/cathode and separator was $5.81 \times 10^7$ and $1.00 \times 10^{-7}$ S m$^{-1}$, respectively. The electrical conductivity of Sn and electrolyte was $9.17 \times 10^5$ and 1.00 S m$^{-1}$, respectively.

## Data availability

The data that support the findings of this study are available from the corresponding author upon reasonable request.

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

## Acknowledgements

This work was financially supported by the Innovation and Technology Commission (ITF Project ITS/029/17), the Key Project for Basic Research of Shenzhen (No. JCYJ20170818104125570), the Research Grant Council of Hong Kong (GRF project: 15301220), the Hong Kong Polytechnic University (ZVRP, ZVGH), and Guangdong-Hong Kong-Macao Joint Laboratory (No. 2019B121205001).

## Author contributions

Z.H. and Y.G. contributed equally to the paper. B.Z. and Z.H. conceived the idea and designed the experiments. Z.H. performed the experiments. Y.G. carried the theoretical simulations. B.Z., Z.H., and H.T. analyzed the data. B.Z., Z.H., and Y.G. prepared this manuscript with inputs from all other coauthors. All authors have given approval to the final version of the manuscript.

## Competing interests

The authors declare no competing interests.
