## [Peer Review File · Nature Communications]

Reviewer #1 (Remarks to the Author):

In order to stabilize both Zn and Na metal anodes at rigorous conditions, this work shows a method to regulate the separator's interfacial chemistry through Sn coating. The symmetric battery constructed by Sn-coated separator processes relative superior cycling stability at a high current density. However, the electrochemical performances of the symmetric battery and the full batteries are general, while no novel preparation technique and profound scientific problems are raised. I cannot support its publication in Nature Communications. Before the authors submit the manuscript to other journals, the following suggestions should be considered.

- (1) The authors claimed that the Sn-coated separator enhance the performance of the Zn battery. The authors should discuss the reason why it influences the reversible Zn plating/stripping processes.
- (2) The surface oxidation of metal separator in aqueous rechargeable battery should be considered in the process of charge and discharge. The XPS and Raman spectrum of the modified separator are needed.
- (3) Since the mixed solution of ZnSO₄ and MnSO₄ was used as the electrolyte in full batteries, more experiments in terms of Zn/Zn batteries with Sn-coated separator should be carried out, and discussion about the electrolyte effect should be given.
- (4) Have the authors ever tested the Zn and Na metal anodes with Sn-coated separator at low current density?
- (5) In Figs 5c, d, the Zn/Zn cells with Sn-coated separator experienced the irregular voltage fluctuation. The authors should provide more stable data to support their conclusion.
- (6) In Fig 5f, the capacity of the full batteries experienced a sharp increase at first 10 cycles, from 50 mAh g⁻¹ to 200 mAh g⁻¹, and then the capacity increases from 200 mAh g⁻¹ to 300 mAh g⁻¹ in the next 50 cycles. This phenomenon is quite abnormal and unconvincing.
- (7) In Supplementary Fig. 12, in the last 30 cycles, the capacity of the full batteries with Sn-coated separator increases from 260 mAh g⁻¹ to 375 mAh g⁻¹. As we all know, MnO₂ is a usual cathode material for aqueous zinc-ion batteries. Many reports have shown that MnO₂ has very excellent electrochemical performance. However, the full battery performance is unstable in this work. The authors should refine their test.

Reviewer #2 (Remarks to the Author):

Dendrites growth is a long-standing issue for the plating/stripping process of metal anodes, such as alkali metal and zinc anodes, which significantly impedes their practical application. This manuscript demonstrates a novel Sn-modified separator for zinc/sodium metal anodes to realize the long-term cycling at simultaneous high current density and charge/discharge capacity, outperforming the performances of previous works. The Sn element is chosen due to its decent stability and good zincophilicity originated from the electrochemical alloy reaction between Sn and Zn. Theoretical calculations and experimental results indicate that the improved plating/stripping behavior comes from the even electric field distribution and face-to-face Zn deposition. This research presents a thoroughly detailed study, providing an exciting approach to suppress metal dendrites growth. Thus, I recommend this manuscript to be published after addressing a few issues listed as follows.

1. The authors prepared the metal modified Ti foils that are used to test the nucleation overpotential of various metal elements by mixing metal powders, Super P, and PVDF (8:1:1 of weight ratio). The

nucleation overpotential of pure Super P modified Ti foils should be supplied to serve as the control group.

2. The focus of this manuscript is to realize the stable cycling at high current density and cycling capacity. However, there are insufficient discussions to explain the disadvantages and failure mechanism at such a rigorous test condition. Authors should provide more discussions in the introduction and Figure 5 on this aspect.

3. In Figures 4a and b, the morphologies of Zn deposition using a pristine separator are nonuniform. What is the reason behind this phenomenon?

4. Why the overpotential decreases with the cycles in Figure 5, especially for Sn coated separator?

5. In Figure 5f, why the CEs of full cells using Sn coated separator outperform those of pristine separator? Moreover, please explain the reasons for the decreased CE of the cells using a pristine separator as the discharge capacity drops.

Reviewer #3 (Remarks to the Author):

This manuscript reports a concept of preventing dendrite formation in Zn and Na batteries by introducing an Sn-coated separator, leading to a uniform plating of Zn (or Na) between the separator and metal anode, thus enabling high capacity and high-rate performance. The manuscript is easy to read and delivers the concept and results clearly. It will give inspiring ideas to researchers in these areas to develop Zn or Na batteries further. However, there are a few issues to be cleared before it is considered for publication. My comments are as follows.

Strictly speaking, though the concept the authors wanted to deliver is obvious, the figure in the middle of Figure 1d is not right in the sense that Zn cannot be plated on the points without an electronic connection to the current collector. The coated Sn particles must directly contact the Zn anode or indirectly through the coated Sn network on the separator's surface. The middle figure shows no contact with the Zn anode, and the particles on the separator seem to be separated from each other.

Fig 2a and Fig 3d show that the stable Zn-Sn alloying (or Zn plating) potential is around -0.02 V. In contrast, Fig 2b shows the reduction potential is just below 0.3 V, which might be explained as a Zn-Sn alloy-formation reaction. The question is why the two voltages are so different.

Fig 2c shows the capacity of 21 mAh/g. Based on the gram of what material was it calculated? I suppose it is Sn, which should be explicitly noted in the text.

Fig 5b. Even for the Sn-coated separator case, there is fluctuation during cycling? Does it imply a micro short-circuit?

Fig 5f. Why the capacity for the Sn-coated separator 10:1 case (for instance) gradually increased up to 60 cycles?

Typos:

Page 4: to guild  build (?)

Page 5: voltage dip  tip

Author response to Reviewers' Comments:

The authors appreciate very much for the invaluable comments provided by the reviewers. All comments are now incorporated in the revision, and the summary is presented in the following of how the amendments are made according to the individual comments.

Reviewer #1

Comments to the Author

*In order to stabilize both Zn and Na metal anodes at rigorous conditions, this work shows a method to regulate the separator's interfacial chemistry through Sn coating. The symmetric battery constructed by Sn-coated separator processes relative superior cycling stability at a high current density. However, **the electrochemical performances of the symmetric battery and the full batteries are general, while no novel preparation technique and profound scientific problems are raised.** I cannot support its publication in Nature Communications. Before the authors submit the manuscript to other journals, the following suggestions should be considered.*

Response: Thanks for the critical comments.

To emphasize the significance and scientific soundness of our strategy, we have carefully revised the manuscript (highlighted in blue) in response to your valuable and insightful suggestions and have done additional experiments to better consolidate our viewpoints. With the clarification of these issues which we did not address well in the previous version, we hope that you would reconsider the publication of this work.

a. The electrochemical performance of the symmetric battery

First, to demonstrate the excellent electrochemical performance of the symmetric battery using the Sn-coated separator, the cycle life of our work and previous reports is summarized in **updated Table S1**. It is worth noting that the plating/stripping life at 2 mA/cm² and 2 mAh/cm² is updated from 2000 h to 3800 h (**updated Figure 5a**), since the cells were still in progress at the

time of initial submission of this manuscript. Besides, as suggested by the reviewer, the cell with Sn-coated separator is also tested at a low current density of 1 mA/cm² and a cycling capacity of 1 mAh/cm². The cell can operate stably for more than 4500 h without any degradation (**new Figure S12**), which is the longest lifetime achieved in Zn metal anodes at same testing condition until now.

Generally, the lifetime of symmetric batteries becomes shorter as the current density and cycling capacity increase¹. Therefore, a detailed comparison between our work and other research on the electrochemical performance of symmetric batteries is performed under three subcategories of test conditions:

(1) The lifetime of our work (3800 h at 2 mA/cm² and 2 mAh/cm²) is substantially better than that of the polyacrylonitrile separator² (350 h at 0.5 mA/cm² and 0.25 mAh/cm²), the Al₂O₃-coated Zn³ (500 h at 1 mA/cm² and 1 mAh/cm²) and the indium-coated Zn⁴ (500 h at 1 mA/cm² and 1 mAh/cm²).

(2) At 5 mA/cm² and 5 mAh/cm², the Sn-coated separator enables steady cycling for 1000 h (**updated Figure 5b**), which is far superior to the vertical graphene-coated separator⁵ (75 h at 5 mA/cm² and 5 mAh/cm²) and the ZnO-coated Zn⁶ (500 h at 5 mA/cm² and 1.25 mAh/cm²).

(3) Even at a high current density of 10 mA/cm² and a cycling capacity of 10 mAh/cm², a stable cycle life of 500 h is achieved for the Sn-coated separator (**updated Figure 5c**), which significantly excels the reduced graphene oxide-coated Zn⁷ (200 h at 10 mA/cm² and 2 mAh/cm²), PVDF/TiO₂-coated Zn⁸ (250 h at 8.85 mA/cm² and 8.85 mAh/cm²) and polyamide-coated Zn (80 h at 10 mA/cm² and 10 mAh/cm²).

In summary, symmetric batteries using Sn-coated separator have absolutely leading cycling stability at multiple current densities and cycling capacities.

To better compare the cycling performance of symmetrical cells at different test conditions, the cumulative capacity⁹ (current density × cycle life) versus the per-cycle areal capacity is plotted in **updated Figure 5d** (based on the data summarized in **updated Table S1**). **Updated Figure 5d** contains the three key parameters of cycle life, current density and cycling capacity (per-cycle areal capacity), which can provide a comprehensive picture of the electrochemical performance of symmetric batteries. This work delivers a high cumulative capacity of 7600 mAh/cm² (2 mA/cm²)

at a cycling capacity of 2 mAh/cm², surpassing the value in most previous works. Moreover, a cumulative capacity of more than 5000 mAh/cm² at 5 mA/cm² and 10 mA/cm² is respectively realized at high cycling capacity of 5 mAh/cm² and 10 mAh/cm². Notably, such high cumulative capacities cannot be achieved at simultaneously high current density and large per-cycle areal capacity in previously reported approaches. For example, the vertical graphene-coated separator⁵ (*Adv. Mater.*, 2020, 32, 2003425) allows a high cumulative capacity of 6000 mAh/cm² at 10 mA/cm² and 1 mAh/cm², but it shows a much lower cumulative capacity of 375 mAh/cm² at 5 mA/cm² and 5 mAh/cm² (only 7.5% of our work). Coincidentally, similar phenomenon is observed on polyamide-coated Zn¹⁰ (*Energy Environ. Sci.*, 2019, 12, 1938-1949): a high cumulative capacity of 4000 mAh/cm² at 0.5 mA/cm² and 0.25 mAh/cm², but an unsatisfying cumulative capacity of 800 mAh/cm² at 10 mA/cm² and 10 mAh/cm² (only 16% of our work). These results highly demonstrate the great superiority of our technique over other methods at a wide range of current densities and cycling capacities.

(Updated) Table S1 Summary of electrochemical performance of Zn plating/stripping behavior through modifying interfacial layer.

Interfacial layer	Current density (mA/cm ²)	Cycling capacity (mAh/cm ²)	Cycle life (h)	Cumulative capacity (mAh/cm ²)	Ref.
	1	1	4500	4500	
Sn-coated separator	2	2	3800	7600	This work
	5	5	1000	5000	
	10	10	500	5000	
Vertical graphene-coated separator	5	5	75	375	5
	10	1	600	6000	
Polyacrylonitrile separator	0.5	0.25	350	175	2
Reduced graphene oxide-coated Zn	10	2	200	2000	7

Carbon-coated Zn	10	1	100	1000	11
PVDF/TiO ₂ - coated Zn	8.85	8.85	250	2212.5	8
ZnO-coated Zn	5	1.25	500	2500	6
Polyamide-coated Zn	0.5 10	0.25 10	8000 80	4000 800	10
Kaolin-coated Zn	4.4	1.1	800	3520	12
Al ₂ O ₃ -coated Zn	1	1	500	500	3
TiO ₂ -coated Zn	1	1	150	150	13
Indium-coated Zn	1	1	500	500	4
MOF-coated Zn	0.5	0.5	700	350	14
CaCO ₃ -coated Zn	0.25	0.05	836	209	15
(0 0 1) facet- TiO ₂ -coated Zn	1	1	460	460	16

(Updated) Figure 5 The electrochemical performance of Zn metal batteries. The cycling performance of Zn/Zn cells using pristine separator and Sn-coated separator tested at **(a)** 2 mA/cm² and 2 mAh/cm², **(b)** 5 mA/cm² and 5 mAh/cm² and **(c)** 10 mA/cm² and 10 mAh/cm². **(d)** Comparison of cycling performance (cumulative capacity versus per-cycle areal capacity) in this work and previously reported works. The electrochemical performance of Zn||MnO₂ batteries using pristine separator and the Sn-coated separator: **(e)** CV at a scan rate of 0.1 mV/s (second cycle); **(f)** Cycling performance at 0.3 A/g, with Zn foil as anode.

(New) **Figure S12** The cycling performance of Zn/Zn cells using pristine separator and Sn-coated separator tested at 1 mA/cm^2 and 1 mAh/cm^2 .

b. The electrochemical performance of the full cells

When it comes to the performance of full cells, the reviewer kindly points out that during the first 10 cycles, a rapid increase in capacity (from 50 mAh/g to 200 mAh/g) is observed on cells using both the pristine separator and the Sn-coated separator (N:P=10:1). Considering that the full cells with Zn foil as the anode (**original Figure S13**) also show an obvious capacity increase in the first 10 cycles, this phenomenon should be ascribed to the instability of prepared MnO_2 cathode rather than the anode. The previously used MnO_2 cathode material is fabricated using the hydrothermal method¹⁷. In order to improve the quality of MnO_2 cathode, several other preparation approaches have been tried, and the most effective one¹⁸ is detailed below:

“2.5 mL of 1.0 M H_2SO_4 and 158 mg of KMnO_4 are mixed with deionized water (30 mL) under magnetic stirring until dissolved. Subsequently, 95 mg of zinc powders are added to the solution and stirred magnetically at $60 \text{ }^\circ\text{C}$ for 6 h. The product is then collected by filtration and washed with deionized water until the pH value is higher than 6. Finally, the product is dried at $80 \text{ }^\circ\text{C}$ for 12 h.

*The phase structure and microstructure morphology of as-received materials are characterized by XRD and SEM, respectively. As shown in the **new Figure S25a**, the*

*diffraction peaks are consistent with the birnessite-type MnO₂ with a layered structure (JCPDS#43-1456)^{18,19}. SEM images show that the morphology of such MnO₂ is flower-like nanospheres (new **Figures S25b, c**), which agrees with the literature¹⁸.”*

Subsequently, the performance of full cells is re-collected using the newly prepared MnO₂ cathode. It is found that the dramatical increase in capacity during the initial 10 cycles is greatly alleviated (**updated Figure 5f**). The full cell using the Sn-coated separator is firstly compared with that using the pristine separator, where the former shows an overwhelming advantage. Specifically, when using Zn foil as the anode, full cell with the Sn-coated separator presents a stable cycle life with a discharge capacity of ~200 mAh/g after 600 cycles. Turning to one with the pristine separator, the discharge capacity is ~159 mAh/g after 130 cycles and then gradually drops to ~94 mAh/g after 260 cycles.

Using a pre-set amount of Zn as the anode, the cycling performance of the full cell is further evaluated at the specific negative-to-positive electrode capacity (N:P) ratios of 10:1. The discharge capacity of the full cell with the pristine separator deteriorates rapidly after about 60 cycles, with a capacity of merely ~63 mAh/g after 80 cycles (**updated Figure S27**), demonstrating the poor Zn reversibility. On the contrary, the highly improved cycle stability is achieved for over 180 cycles (discharge capacity of ~145 mAh/g) on the full cell with the Sn-coated separator.

The CV curves of full cells using the pristine separator and the Sn-coated separator are compared in the **updated Figure 5e**. They present the same Mn-ion redox peaks, which is in good agreement with the previous work¹⁸. The full cell with the Sn-coated separator shows a lower oxidation potential, a higher reduction potential, and a larger peak current than that with the pristine separator, suggesting the improved reaction kinetics for the Sn-coated separator^{12,20,21}. This conclusion is further verified by the rate performance of full cells (**updated Figure S26**). At a high current density of 0.75 A/g (equivalent to ~2.5 C), the discharge capacity of the full cell with Sn-coated separator doubles that with the pristine separator (107 mAh/g vs. 53 mAh/g).

To show the decent stability of full cells using the Sn-coated separator, the performance of our work and other research is summarized in **new Table S3**. A discharge capacity of ~200 mAh/g and a capacity retention of ~100% are realized using the Sn-coated separator after over 600 cycles, which is competitive with those previous reports.

The above data shows that (1) the performance of full cell using the Sn-coated separator is much superior to that using the pristine separator with the same cathode and anode; (2) the stability of full cell using the Sn-coated separator is comparable to the reported research. (3) the quality of the MnO₂ cathode greatly affects the performance of the full cells.

It is well known that different cathode types (e.g., MnO₂, V₂O₅ and NVP), as well as different preparation methods for the same type of cathode, can greatly influence the electrochemical performance of full cells. Since our work is not focused on the cathode preparation, we use the commercially available cathode (activated carbon (AC)) to further compare the full cell performance with other literature that also use commercial AC, which will provide a more realistic picture of the performance enhancement of our Sn-coated separator.

The SEM images of the prepared AC electrode (**new Figure S28a**) display that the AC particles have irregular shapes, which is in good agreement with the reported results²². The performance of Zn/AC full cells using Sn-coated separator is tested at 12 A/g. Surprisingly, such cell can stably deliver a discharge capacity of 54 mAh/g for more than 20000 cycles (**new Figure S28b**). **New Table S4** summaries the electrochemical performance of our work and other reported full cells using AC as cathode. Both current density and cycle number are found to be significantly higher than those studies, demonstrating the great superiority of our method. Furthermore, **new Figure S28c** presents more clearly the excellent stability of full cells using Sn-coated separator, even at an ultrahigh current density, which is consistent with the results of Zn/Zn batteries (**updated Figure 5d**).

In the future, we will further optimize the synthesis technique of MnO₂ cathodes and fabricate other cathode materials to match the highly stable Zn metal anode achieved by the Sn-coated separator. Currently, our work is mainly focused on improving the stability of Zn metal anodes under stringent test conditions (with both high current density and large cycling capacity), which is a problem not addressed in most reports.

(Original) Figure S13 The cycle performances of Zn||MnO₂ batteries using pristine separator and Sn-coated separator at 0.3 A/g, with Zn foil as anode.

(New) Figure S25 (a) The XRD spectrum and (b, c) SEM images of newly prepared MnO₂ materials.

(Updated) Figure S27 The cycling performances of Zn||MnO₂ batteries using pristine separator and Sn-coated separator at 0.3 A/g with N:P ratios of 10:1

(Updated) Figure S26 The rate performances of Zn||MnO₂ batteries using pristine separator and Sn-coated separator.

(New) Table S3 Summary of electrochemical performance of Zn metal batteries through modifying interfacial layer.

Modification	Cathode	Discharge capacity (mAh/cm ²)	Cycle Number	Capacity retention (%)	Ref.
Sn-coated separator	MnO₂	~200	600	~100	This work
Vertical graphene-coated separator	V ₂ O ₅	~150	1000	~75	5
Polyacrylonitrile separator	None				2
Reduced graphene oxide-coated Zn	V ₃ O ₇ ·H ₂ O	~200	1000	~79	7
Carbon-coated Zn	NVP	~70	1000	~65	11

PVDF/TiO ₂ -coated Zn	MnO ₂	~234	300	~100	8
ZnO-coated Zn	MnO ₂	~212.9	500	~100	6
Polyamide-coated Zn	MnO ₂	~176.1	1000	~88	10
Kaolin-coated Zn	MnO ₂	~190	600	~86	12
Al ₂ O ₃ -coated Zn	MnO ₂	~250	200	~74	3
TiO ₂ -coated Zn	MnO ₂	~150	1000	~85	13
CaCO ₃ -coated Zn	MnO ₂	~185	1000	~86	15
(0 0 1) facet-TiO ₂ -coated Zn	MnO ₂	~80	300	~84	16

(New) **Figure S28** (a) SEM images of AC electrode. (b) cycling performance of Zn/AC full cells using Sn-coated separator at 12 A/g. (c) Comparison of cycling performance (cycle number versus current density) in this work and previously reported works.

(New) **Table S4** Summary of electrochemical performance of Zn metal batteries using AC as cathode.

Modification	Current density (A/g)	Discharge capacity (mAh/cm ²)	Cycle Number	Capacity retention (%)	Ref.
Sn-coated separator	12	~54	20000	~100	This work
Vertical graphene-coated separator	5	~60	5000	~94	5
Indium-coated Zn	2	~70	5000	~86	4
MOF host	4	~58	20000	~72	23
Zincic perfluorinated sulfonic acid membrane	0.2	~138	50	None	24
Konjac glucomannan-coated Zn	1	~50	5000	~98.8	25

c. Preparation technique

Although there are many different film-preparation techniques, magnetron sputtering is easy to handle, reproducible, highly uniform and can precisely control the thickness of the film at the nanoscale level²⁶. These characteristics give it unparalleled advantages for separator modification. We first examine the thickness controllability of magnetron sputtering by characterizing the thickness of films generated on the silicon wafer at different sputtering times. The cycling stability

of cells assembled with separators having different thickness of sputtered Sn layers is also compared to investigate the necessity of thickness control to the battery performance. Secondly, performance of cells using separators modified by other techniques is evaluated and compared.

The sputtering time is set as 0.5, 1, 2 and 5 min, respectively. The corresponding film thickness at different sputtering times are evaluated using atomic force microscope (AFM). As shown in **new Figure S7**, the thickness after 0.5 min sputtering time is ~ 20 nm, followed by ~ 40 nm (1 minute), ~ 80 nm (2 min) and ~ 170 nm (5 min). A linear relationship between film thickness and sputtering time is found in the sputtering time range of 0-5 min, indicating that the film thickness can be precisely tuned using magnetron sputtering.

Actually, the controllable coating thickness plays a critical role in achieving stable Zn plating/stripping under rigorous testing condition. We assess the stability of Zn metal anodes using Sn-coated separators at different sputtering times. These samples are denoted as Sn-coated separator-X, where X represents the sputtering time (min). SEM images shows that bigger Sn particles and denser Sn layer are generated as the sputtering time increases (**new Figure S20**). As shown in **new Figure S21**, all Zn/Zn cells with these Sn-coated separators exhibit highly improved lifetime compared with the pristine separator at 5 mA/cm^2 and 5 mAh/cm^2 . At the more rigorous test condition of 10 mA/cm^2 and 10 mAh/cm^2 , the cycle life of Sn-coated separator-0.5 and Sn-coated separator-5 is much lower than that of Sn-coated separator-1 and Sn-coated separator-2 samples. These data suggest an optimal thickness of Sn layer for the separator modification (see **Part “d”** for details) to maximize the synergistic effect of the even electric field and face-to-face growth of Zn. Fortunately, *the magnetron sputtering method allows us to precisely tune the coating thickness at the nanoscale level*, enabling significant improvements in cell's stability especially under stringent test conditions.

To demonstrate the superiority of this modification technique, the Sn-coated separator and Sn-coated Zn foil are also prepared using the doctor blading technique²⁷, which is the most frequently used approach to prepare electrodes in commercial lithium ion batteries. This technique can produce wet films with thicknesses range from 20 to several hundred microns. Firstly, the slurry is prepared by uniformly mixing 90 wt% Sn nanopowders (<150 nm particle size), and 10 wt% polyvinylidene difluoride (PVDF) in N-methyl-2-pyrrolidone solvent. The homogeneously mixed slurry is then applied to the cellulose separator and the Zn foil using the doctor blade method

with the minimum thickness grade for the scraper (25 μm) to make the Sn-coated separator and Sn-coated Zn foil, respectively. For convenience, the samples are marked as DSn-coated separator and DSn-coated Zn foil. As shown in **new Figures 23a-d**, aggregation and non-uniform distribution of Sn nanoparticles are observed on both the DSn-coated separator and DSn-coated Zn foil. Besides, the separator and Zn foil are completely covered by the Sn layer due to their micron thicknesses of $\sim 12 \mu\text{m}$ (**new Figures 23e, f**).

The cycling performance of Zn/Zn cells with Sn-coated and DSn-coated separators is compared at 5 mA/cm^2 and 5 mAh/cm^2 . The cell using the DSn-coated separator shows an enormous increase in voltage potential after only $\sim 40 \text{ h}$ (**new Figure R1**), which is even worse than that using the pristine separator. By contrast, stable cycle life of 1000 h is achieved for the cell using the Sn-coated separator under the same condition. The much inferior performance of DSn-coated separator than the pristine separator should stem from the uneven distributed Sn on the separator that causes nonuniform Zn deposition. In addition, although we have chosen the minimum thickness grade for the scraper, the Sn layer on the DSn-coated separator still has a thickness of around $12 \mu\text{m}$. The thick Sn layer probably hinders the transport of Zn^{2+} and results in an elevated overpotential.

To verify the importance of separator modification, we also investigate the influences of modifying the Zn foil with a Sn coating both by magnetron sputtering and doctor blade method. As shown in **new Figure R1**, for the case of magnetron sputtering, the performance of battery with modified Zn foil are inferior to those with modified separator. In addition, for cells both with modified Zn foil, the one prepared by magnetron sputter shows better performance than that using the doctor blade.

All these results demonstrate that using the technique of magnetron sputtering to modify the separator allows precise tuning of the coating thickness at the nanometer level, which maximizes the cell performance at both high current densities and large cycling capacities.

(New) Figure S7 AFM images of Sn-coated separator with different sputtering time, **(a, e)** 0.5 minute, **(b, f)** 1 minute, **(c, g)** 2 minutes and **(d, h)** 5 minutes. **(i)** The thickness and **(j)** average roughness of Sn-coated separator with different sputtering time.

(New) Figure S20 SEM images of **(a)** Sn-coated separator-0.5, **(b)** Sn-coated separator-1, **(c)** Sn-coated separator-2 and **(d)** Sn-coated separator-5.

(New) Figure S21 The cycling performances of cells using Sn-coated separator-X tested at 5 mA/cm² and 5 mAh/cm².

(New) Figure S23 SEM images of (a, b) DSn-coated Zn foil and (c, d) DSn-coated separator. Cross section images of (e, f) DSn-coated Zn foil.

(New) Figure R1 The cycling performances of cells using Sn-coated separator, Sn-coated Zn foil, DSn-coated separator, and DSn-coated Zn foil tested at 5 mA/cm^2 and 5 mAh/cm^2 .

d. Profound scientific problems

To elucidate the scientific advancement of this work, the working mechanism of proposed methodology is compared with different methods of stabilizing Zn metal anodes through modifying the interfacial layer. Moreover, additional experiments are carried out to clarify the universality and superiority of this working mechanism.

Most previous strategies for performance optimization focus on reducing local current density or homogenizing the Zn^{2+} flux. For example, the introduction of vertical graphene⁵ into the separator effectively homogenized the electric field distribution and Zn^{2+} flux, achieving an improved cycle life of $\sim 80 \text{ h}$ (5 mA/cm^2 , 5 mAh/cm^2). Similarly, kaolin-coated Zn anode¹² was proposed to enable homogeneous Zn^{2+} migration and a lifetime of $\sim 800 \text{ h}$ (4.4 mA/cm^2 , 1 mAh/cm^2). However, these strategies can only somewhat slow down the generation of dendrites and improve the cell performance to a limited extent, as Zn dendrites formation is thermodynamically and kinetically favorable^{28, 29}, especially at rigorous testing conditions.

Unlike previously reported strategies, the Sn-coated separator we used not only homogenizes the electric field distribution^{30,31} (**updated Figures 1a, b**), but also utilizes the zincophilicity of Sn to enable the face-to-face Zn growth between anode and separator^{32,33} (**original Figure 4**), thus greatly curbing the dendrite hazard (**updated Figures 1c, d**). With the synergistic effects of these

two mechanisms, we achieve a cumulative capacity of more than 5000 mAh/cm² (1000 h at 5 mA/cm² for 5 mAh/cm² and 500 h at 10 mA/cm² for 10 mAh/cm²), which is in the leading position under different test conditions (see **Part “a”** for performance comparison). In the previous submitted version, we extend this strategy to Na metal anodes and achieve a highly prolonged lifetime of 450 h at a high current density (5 mA/cm²) and cycling capacity (5 mAh/cm²) (**original Figure 6b**). In the revised version, we further apply this strategy to potassium (K) metal anodes and realize a lifetime of 450 h at a large current density (3 mA/cm²) and cycling capacity (3 mAh/cm²), as seen from **new Figure S31**. These overwhelming performances *demonstrate the universality and superiority of our strategy* in solving the nuisance caused by dendrite growth that afflicting alkaline metal batteries.

To illustrate the working mechanism more clearly, we have added a series of comparative experiments. We prepared Sn-coated Zn foils (sputtering time: 1 min), Ag-coated separators (sputtering time: 1 min), and Sn-coated separators (sputtering time: 0.5, 2, and 5 min) using magnetron sputtering method. The electrochemical performance of Zn/Zn symmetric cells using these samples is evaluated.

As shown in **new Figure S17**, cell using Sn-coated Zn foil shows a slightly improved cycle life compared with that using pristine Zn foil, but it is much shorter than that with Sn-coated separator. Although a zincophilic coating on the substrate is reported to regulate Zn nucleation and leads to relatively uniform Zn deposition. Such enhancement is generally limited to mild testing conditions. For example, Zn foil decorated with zincophilic gold (Au)³⁴ achieved a lifetime of 2000 h at a very low current density and cycling capacity (0.25 mA/cm² and 0.05 mAh/cm²). **New Figure S18** clearly displays the deposition/stripping process of Zn on Sn-coated Zn foil. Specifically, the zincophilic Sn coating on Zn foil facilitates more uniform Zn deposition during initial stages. However, under severe test conditions (high current and high cycling capacity), the uneven Zn²⁺ flux induced by the non-uniform electric field distribution greatly counteracts the effect of zincophilic coating, which triggers uneven Zn growth and leads to rapid short circuit after repeated cycles. For the case of Sn-coated separator, the Zn²⁺ flux reaches the electrode surface in an orderly manner driven by the uniform electric field, thus remaining effective at high currents and high cycling capacities.

In the initially submitted manuscript, in order to screen the ideal metal elements, we compare the zincophilicity of various metals using the parameter nucleation overpotential (η). Among Sn, Ag, Bi, and Sb, Ag is found to have the second highest Zn affinity after Sn (**original Figure S4**). Therefore, we prepare Ag-coated separator to verify the importance of zincophilicity for coating. As shown in **new Figure S19**, the lifetime of the Zn/Zn cell with the Ag-coated separator is extended from 170 h to 460 h compared to the cell using pristine separator, while a stable cycle life of 1000 h is observed on cell using Sn-coated separator. Notably, a smaller overpotential is observed for the Ag-coated separator compared with the Sn-coated separator, probably because Ag has the highest electric conductivity among all the candidate metals³⁵ (about an order of magnitude higher than Sn). This result demonstrates firstly that the zincophilic separator coating provides improved cycling performance of the battery and secondly that the degree of improvement in cell performance is highly correlated with the degree of zincophilicity of the coating.

As discussed in **Part “c”** above, the coating thickness plays a critical role in achieving stable Zn plating/stripping under rigorous testing conditions. Among sputtering times of 0.5, 1, 2 and 5 min, the best stability is achieved with Sn-coated separator-1. For Sn-coated separator-0.5, the Sn layer is too thin to realize even electric field distribution and face-to-face growth of Zn, resulting in inferior electrochemical performance. It is observed that batteries using Sn-coated separator-2 and Sn-coated separator-5 exhibit relatively poor cycle life compared to the case with Sn-coated separator-1 (but still far superior to the case using pristine separator). Similar overpotentials are observed on these cells, indicating that the variation in thickness of nanoscale Sn layer has a negligible effect on the transport of Zn^{2+} through the modified separator. Namely, the difference of Zn^{2+} transport is not responsible for the shortened lifetimes of cells using Sn-coated separator-2 and Sn-coated separator-5. The 5 min sputtering time makes the Sn layer on the separator a continuous whole, which is indicated by the small resistance value between two points in the coating (**new Figure S22**). This means that the Sn-coated separator-5 will have only one function of achieving face-to-face growth of Zn without the capability of homogenizing the electric field distribution, thus losing the competition when compared with Sn-coated separator-1 with two synergetic mechanisms. These results demonstrate that precise tuning of the Sn coating thickness at the nanoscale level to maximize the synergistic effect of these two mechanisms (equipotential surface and face-to-face Zn growth) is necessary to achieve significant improvements in stability, especially under stringent test conditions.

In summary, we first propose that Sn-coated separator has two benefits for stabilizing Zn metal anodes: the equipotential surface delays the initiation of Zn dendrites and the face-to-face growth of Zn eliminates the inevitably formed Zn dendrites. Additional experiments further confirm the significance of zincophilicity and the unique synergy of these two working mechanisms. By comparison with the performances of cells using Zn foil modified by Sn layer and previous reports, it is found that our proposed mechanism possesses a remarkable superiority and advancement. In addition, this method with decent universality would be readily extended to Na/K metal anodes and achieve highly prolonged lifetimes.

(Updated Figure 1) Theoretical calculation and protection mechanism of modified separator.
(a) Electric field distribution with the pristine separator. **(b)** Electric field distribution in the non-contact region of the modified separator and the anode. **(c)** Schematic illustration of Zn deposition with the pristine separator. **(d)** Schematic illustration of Zn deposition in the contact region of the modified separator and the anode.

(Original) Figure 4 SEM images of the electrodes after Zn deposition at 1 mA/cm². Zn deposition on Ti current collector using pristine separator with a cycling capacity of **(a)** 1 mAh/cm² and **(b)** 4 mAh/cm². Zn deposition on Ti current collector using Sn-coated separator with a cycling capacity of **(c)** 1 mAh/cm² and **(d)** 4 mAh/cm². Zn deposition on Sn-coated separator with a cycling capacity of **(e)** 1 mAh/cm² and **(f)** 4 mAh/cm².

(Original) Figure 6 The electrochemical performances of Na metal batteries. The cycling performances of Na/Na cells using pristine separator and Sn-coated separator are tested at **(a)** 2 mA/cm² and 2 mAh/cm², **(b)** 5 mA/cm² and 5 mAh/cm². **(c)** The cycling performances of Na||NVPF batteries using pristine separator and Sn-coated separator at 0.1 A/g, with N:P ratios of 5:1.

(New) **Figure S31** The cycling performances of K/K cells using pristine separator and Sn-coated separator are tested at 3 mA/cm^2 and 3 mAh/cm^2 .

(New) **Figure S17** The cycling performances of cells using Sn-coated separator and Sn-coated Zn foil tested at 5 mA/cm^2 and 5 mAh/cm^2 .

(New) **Figure S18** Schematic illustration of Zn deposition with Sn-coated Zn foil.

(Original) **Figure S4** The nucleation overpotential of Ti and metal-Ti current collectors.

(New) Figure S19 The cycling performances of cells using Sn-coated separator and Ag-coated separator tested at 5 mA/cm² and 5 mAh/cm².

(New) Figure S22 The resistance of Sn-coated separator-5.

(1) The authors claimed that the Sn-coated separator enhance the performance of the Zn battery. The authors should discuss the reason why it influences the reversible Zn plating/stripping processes.

Response: We have included a separate **Part “(d) Profound scientific problems”** in the previous section detailing the working mechanism of Sn-coated separator. Here, to display the influence of Sn-coated separator for Zn reversibility more directly, we perform CE tests of Ti/Zn cells using pristine separator and Sn-coated separator at 5 mA/cm² for 5 mAh/cm² and 10 mA/cm² for 10 mAh/cm², respectively. CE is a critical parameter to evaluate the reversibility of Zn deposition/stripping, which is calculated based on the capacity ratio of stripping to plating. As shown in **(new) Figures S11**, the cells using Sn-coated separator show high and stable CE values compared with that using pristine separator at both testing conditions. Moreover, a rapid short circuit is observed on the cell using the pristine separator, indicating rampant Zn dendrite growth during the plating/stripping process. Therefore, Sn-coated separator allows for better Zn reversibility and suppresses the growth of Zn dendrites.

As known, the poor reversible Zn plating/stripping is mainly originated from the formation of Zn dendrites with a large surface area, which promotes severe parasitic reactions between Zn metal anode and electrolyte. In addition, Zn dendrites may lose contact with the electrode and become “dead Zn” upon cycling. These would render low reversibility for Zn plating/stripping. As shown in **original Figure 4b**, at a deposition capacity of 4 mAh/cm², loose structures with sharp tips are observed on Ti current collector using the pristine separator, which is not only detrimental to the reversibility of Zn metal anode but also might pierce the separator and cause a short circuit upon cycling. On the contrary, the Ti current collector in the cell using Sn-coated separator presents a greatly improved Zn deposition behavior. Specifically, dense and uniform Zn coatings are obtained (**original Figure 4d**). The corresponding SEM images of Sn-coated separator also present a smooth structure of Zn metal (**original Figure 4f**).

The dendrite-free morphologies can be attributed to two mechanisms as we mentioned previously. Firstly, for the region where the Sn coating is not in contact with the electrode, the equipotential property of Sn homogenizes the electric field distribution between the separator and the electrode (as evidenced by the FEM simulation in **updated Figures 1a, b**), allowing the Zn²⁺ flux to reach the electrode surface uniformly. Secondly, within the region where Sn coating is in contact with Ti current collector, Zn ions will prefer to deposit on Sn due to its zincophilicity instead of on Ti current collector, in turn realizing face-to-face growth of Zn. As shown in **original Figure S11**, both mechanisms will smooth the Zn deposition and the face-to-face grown Zn will merge at

later stages to suppress the dendrite growth. Thus, an improved reversibility and stability of Zn plating/stripping could be achieved by using the Sn-coated separator. The corresponding discussions have been incorporated to the revised manuscript (**Page 10 and 11**) and supporting information (**Page S13 and S14**).

(New) Figure S11 CEs of Ti/Cu cells and detailed deposition/stripping voltage curves using pristine separator and Sn-coated separator at **(a)** 5 mA/cm² and 5 mAh/cm² and **(b)** 10 mA/cm² and 10 mAh/cm².

(Original) Figure 4 SEM images of the electrodes after Zn deposition at 1 mA/cm². SEM images of the electrodes after Zn deposition at 1 mA/cm². Zn deposition on Ti current collector using pristine separator with a cycling capacity of (a) 1 mAh/cm² and (b) 4 mAh/cm²; Zn deposition on Ti current collector using Sn-coated separator with a cycling capacity of (c) 1 mAh/cm² and (d) 4 mAh/cm²; Zn deposition on Sn-coated separator with a cycling capacity of (e) 1 mAh/cm² and (f) 4 mAh/cm².

(Original) Supplementary Figure 11 SEM images of (a) cycled Zn using Sn-coated separator and (b) cycled Zn on Sn-coated separator after 200 cycles at 5 mA/cm² and 5 mAh/cm².

(2) *The surface oxidation of metal separator in aqueous rechargeable battery should be considered in the process of charge and discharge. The XPS and Raman spectrum of the modified separator are needed.*

Response: As suggested by the reviewer, the XPS and Raman spectra of Sn-coated separator after 20 cycles (CSn-coated separator) are collected. As shown in **new Figure S9a**, the peak position of Sn 3d_{5/2} in XPS spectrum shows negligible shift after cycling in the electrolyte, confirming that the Sn layer on the separator is stable in the electrolyte. This is further proved by Raman spectra where no new peaks are observed for the cycled Sn-coated separators compared to the pristine Sn-coated separator (**new Figure S9b**). Therefore, the Sn metal on the separator is highly resistant to oxidation during cycling in the aqueous electrolyte. The corresponding discussions have been added to the revised manuscript (**Page 8**) and supporting information (**Page S11**).

(New) **Figure S9** The (a) XPS and (b) Raman spectra of pristine and after 20-cycle Sn-coated separator.

(3) *Since the mixed solution of ZnSO₄ and MnSO₄ was used as the electrolyte in full batteries, more experiments in terms of Zn/Zn batteries with Sn-coated separator should be carried out, and discussion about the electrolyte effect should be given.*

Response: According to the previous report¹⁷, the additive of MnSO₄ could suppress the dissolution of Mn²⁺ from MnO₂ cathode, thus improving the stability of cathode. Besides, such additive has been commonly adopted in full cell tests by many researchers^{10,12,15,16}. Thus, we introduce 0.2 M MnSO₄ into 2 M ZnSO₄ electrolyte during the test of full cells.

As suggested by the reviewer, the cycling performance of Zn/Zn cells using 2 M ZnSO₄ + 0.2 M MnSO₄ as electrolyte is evaluated at 5 mA/cm² for 5 mAh/cm² and 10 mA/cm² for 10 mAh/cm². As shown in **new Figure S32**, cells using Sn-coated separator with MnSO₄ additive could stably run for 900 h (5 mA/cm² and 5 mAh/cm²) and 500 h (10 mA/cm² and 10 mAh/cm²), which is similar with the case without MnSO₄ additive. Analogously, the stability using pristine separator with/without MnSO₄ additive are also comparable. These results demonstrate that the MnSO₄ additive has a negligible influence on Zn deposition/stripping behavior. The corresponding discussions have been added to the revised manuscript (**Page 20**) and supporting information (**Page S34**).

(New) **Figure S32** The cycling performance of cells using pristine separator and Sn-coated separator in 2 M ZnSO₄ + 0.2 M MnSO₄ electrolyte at (a) 5 mA/cm² and 5 mAh/cm² and (b) 10 mA/cm² and 10 mAh/cm².

(4) Have the authors ever tested the Zn and Na metal anodes with Sn-coated separator at low current density?

Response: As suggested, the Zn/Zn and Na/Na symmetry cells with Sn-coated separator are tested at low current density of 1 mA/cm². As shown in **new Figure S12**, Zn/Zn cell with Sn-coated separator could stably run for more than 4500 h without degradation, which is much longer than that using the pristine separator (~900 h). In addition, a prolonged lifetime of over 1200 h is also realized on the Na/Na symmetry cell using the Sn-coated separator compared to that using the pristine separator (880 h) (**new Figure S30**). These results verify the high applicability of our

strategy at low current density. The corresponding discussions have been added to the revised manuscript (Page 11 and 16) and supporting information (Page S15 and S32).

(New) Figure S12 The cycling performances of Zn/Zn cells using pristine separator and Sn-coated separator tested at 1 mA/cm^2 and 1 mAh/cm^2 .

(New) Figure S30 The cycling performances of Na/Na cells using pristine separator and Sn-coated separator tested at 1 mA/cm^2 and 1 mAh/cm^2 .

(5) In Figs 5c, d, the Zn/Zn cells with Sn-coated separator experienced the irregular voltage fluctuation. The authors should provide more stable data to support their conclusion.

Response: Thank you for pointing this out. The probable reason for this phenomenon is the concurrent deposition of Zn on the anode and the Sn-coated separator. The overpotentials required

for depositing Zn^{2+} on the anode and the Sn-coated separator are different. The deposition amount of Zn on anode and Sn-separator slightly varies with cycles, resulting in various potentials for different cycles. We have re-tested Zn/Zn cells using Sn-coated separator under these two testing conditions at constant temperature and provide two reproducible experimental data for both conditions. As shown in **updated Figure 5** and **new Figure S13**, the Sn-coated separator delivers stable Zn deposition/stripping curves after long-term cycling at 5 mA/cm^2 for 5 mAh/cm^2 and at 10 mA/cm^2 for 10 mAh/cm^2 , indicating its excellent potential for practical applications. The corresponding discussions have been added to the revised manuscript (**Page 11**) and supporting information (**Page S15 and S16**).

(Updated) Figure 5 The electrochemical performance of Zn metal batteries. The cycling performance of Zn/Zn cells using pristine separator and Sn-coated separator tested at (a) 2 mA/cm^2 and 2 mAh/cm^2 , (b) 5 mA/cm^2 and 5 mAh/cm^2 and (c) 10 mA/cm^2 and 10 mAh/cm^2 . (d) Comparison of cycling performance (cumulative capacity versus per-cycle areal capacity) in this work and previously reported works. The electrochemical performance of Zn||MnO₂ batteries

using pristine separator and the Sn-coated separator: (e) CV at a scan rate of 0.1 mV/s (second cycle); (f) Cycling performance at 0.3 A/g, with Zn foil as anode.

(New) **Figure S13** Reproducible experimental data for the cycling performances of Zn/Zn cells using pristine separator and Sn-coated separator tested at (a) 5 mA/cm² and 5 mAh/cm² and (b) 10 mA/cm² and 10 mAh/cm².

(6) In Fig 5f, the capacity of the full batteries experienced a sharp increase at first 10 cycles, from 50 mAh g⁻¹ to 200 mAh g⁻¹, and then the capacity increases from 200 mAh g⁻¹ to 300 mAh g⁻¹ in the next 50 cycles. This phenomenon is quite abnormal and unconvincing.

Response: This phenomenon is ascribed to the fact that the prepared MnO₂ cathodes are not very stable, which is verified by the performance of full cells consisting of Zn foil as anodes, as described in the previous Part “(b) The electrochemical performance of the full cells”. As shown in **original Figure S13**, the capacity of full cells using both pristine and Sn coated separators experiences a continuous increase upon cycling, indicating the instability of prepared MnO₂ cathodes. The MnO₂ cathode material is fabricated using the hydrothermal method¹⁷.

To obtain stable and reliable MnO₂ cathode materials, we turn to employ other method (chemical reaction according to the previous report) and try our utmost effort to synthesize them¹⁸. Concretely, 2.5 mL of 1.0 M H₂SO₄ and 158 mg of KMnO₄ were mixed with deionized water (30 mL) under magnetic stirring until dissolving. Subsequently, 95 mg of zinc powders were added into the solution and were magnetically stirred at 60 °C for 6 h. The production was collected by filtration and washed with deionized water until pH is higher than 6. Finally, it is dried at 80 °C for 12 h. The XRD and SEM are respectively used to observe the phase structure and microstructure morphologies of as-received sample. As shown in **new Figure S25a**, the diffraction peaks are consistent with the birnessite-type MnO₂ with a layered structure (JCPDS#43-1456)^{18,19}.

SEM images show that the morphologies of such MnO₂ is flower-like nanospheres (**new Figures S25b, c**), which agrees with the literature¹⁸.

Subsequently, the performance of full cells is re-collected using this MnO₂ cathode. It is found that the phenomenon of dramatical increased capacity in initial cycles is greatly alleviated. The CV curves of full cells using pristine separator and Sn-coated separator are compared in **updated Figure 5e**. They present the same Mn-ion redox peaks, which agrees well with the previous works¹⁸. The full cell with Sn-coated separator shows a lower oxidation potential, a higher reduction potential and peak current than that with pristine separator, suggesting improved reaction kinetics for Sn-coated separator^{12,20,21}. Supportive evidence could be found at the rate performances of full cells **updated Figure S26**. At a high current density of 0.75 A/g (equivalent to 2 C), a full cell with the Sn-coated separator can provide a discharge capacity equivalent to approximately 200% of that with the pristine separator (107 mAh/g vs. 53 mAh/g).

As shown in **updated Figure 5f**, full cell with the Sn-coated separator presents a stable cycle life with a discharge capacity of ~200 mAh/g after 600 cycles. Turning to one with the pristine separator, the discharge capacity is ~159 mAh/g after 130 cycles and then gradually drops to ~94 mAh/g after 260 cycles. More critically, using a pre-set amount of Zn as the anode, the cycling performance of the full cell is further evaluated at the specific negative-to-positive electrode capacity (N:P) ratios of 10:1. The discharge capacity of the full cell with the pristine separator deteriorates rapidly after about 60 cycles, with a capacity of merely ~63 mAh/g after 80 cycles (**updated Figure S27**), demonstrating the poor Zn reversibility. On the contrary, the highly improved cycle stability is achieved over 180 cycles (discharge capacity of ~145 mAh/g) on the full cell with the Sn-coated separator.

Therefore, the improved stability is realized on full cells using Sn-coated separator after re-preparing the MnO₂ cathode. In the future, we will further optimize the synthesis technique of MnO₂ cathodes and fabricate other cathode materials to match this highly stable Zn metal anode achieved by the Sn-coated separator. The corresponding discussions have been added to the revised manuscript (**Page 14 and 15**) and supporting information (**Page S28 and S29**).

(Original) Figure S13 The cycle performances of Zn||MnO₂ batteries using pristine separator and Sn-coated separator at 0.3 A/g, with Zn foil as anode.

(New) Figure S25 (a) The XRD spectrum and (b, c) SEM images of newly prepared MnO₂ materials.

(Updated) Figure S26 The rate performances of Zn||MnO₂ batteries using pristine separator and Sn-coated separator.

(Updated) Figure S27 The cycling performances of Zn||MnO₂ batteries using pristine separator and Sn-coated separator at 0.3 A/g with N:P ratios of 10:1

(7) In Supplementary Fig. 12, in the last 30 cycles, the capacity of the full batteries with Sn-coated separator increases from 260 mAh g⁻¹ to 375 mAh g⁻¹. As we all know, MnO₂ is a usual cathode material for aqueous zinc-ion batteries. Many reports have shown that MnO₂ has very excellent electrochemical performance. However, the full battery performance is unstable in this work. The authors should refine their test.

Response: Thanks for the reviewer's comments. We have re-fabricated stable and reliable MnO₂ cathode materials by employing alternative methods. As shown in **update Figure 5f and Figure S26**, a substantial improvement in stability is realized using the new MnO₂ cathode. The rate performance using the Sn-coated separator is superior to that using the pristine separator. For example, at a high current density of 0.75 A/g (equivalent to 2 C), a full cell with the Sn-coated separator can provide a discharge capacity equivalent to approximately 200% of that with the pristine separator (107 mAh/g vs. 53 mAh/g).

Moreover, to better reflect the contribution of our separator modification, we additionally prepare full cells with commercial activated carbon (AC) as the cathode. The performance of Zn/AC full cells using Sn-coated separator is tested at 12 A/g. Surprisingly, such cell could stably deliver a discharge capacity of 54 mAh/g over 20000 cycles (**new Figure S28b**). **New Figure S28c** plotted using these two parameters more clearly presents the excellent stability of full cells using

Sn-coated separator even at ultrahigh current density, which is consistent with the results of Zn/Zn battery (**updated Figure 5d**). The corresponding discussions have been added to the revised manuscript (**Page 15**) and supporting information (**Page S29 and S30**).

(New) Figure S28 (a) SEM images of AC electrode. (b) cycling performance of Zn/AC full cells using Sn-coated separator at 12 A/g. (c) Comparison of cycling performance (cycle number versus current density) in this work and previously reported works.

References

1. Yang Q, *et al.* Dendrites in Zn-based batteries. *Adv. Mater.* **32**, e2001854 (2020).
2. Lee BS, *et al.* Dendrite suppression membranes for rechargeable zinc batteries. *ACS Appl. Mater. Interfaces* **10**, 38928-38935 (2018).

3. He H, Tong H, Song X, Song X, Liu J. Highly stable Zn metal anodes enabled by atomic layer deposited Al₂O₃ coating for aqueous zinc-ion batteries. *J. Mater. Chem. A* **8**, 7836-7846 (2020).
4. Han D, *et al.* A corrosion-resistant and dendrite-free zinc metal anode in aqueous systems. *Small* **16**, e2001736 (2020).
5. Li C, *et al.* Directly grown vertical graphene carpets as janus separators toward stabilized Zn metal anodes. *Adv. Mater.* **32**, e2003425 (2020).
6. Xie X, *et al.* Manipulating the ion-transfer kinetics and interface stability for high-performance zinc metal anodes. *Energy Environ. Sci.* **13**, 503-510 (2020).
7. Shen C, *et al.* Graphene-boosted, high-performance aqueous Zn-ion battery. *ACS Appl. Mater. Interfaces* **10**, 25446-25453 (2018).
8. Zhao R, *et al.* Redirected Zn electrodeposition by an anti-corrosion elastic constraint for highly reversible Zn anodes. *Adv. Funct. Mater.* **31**, 2001867 (2020).
9. Ma L, *et al.* Realizing high zinc reversibility in rechargeable batteries. *Nat. Energy*, **5**, 743-749 (2020).
10. Zhao Z, *et al.* Long-life and deeply rechargeable aqueous Zn anodes enabled by a multifunctional brightener-inspired interphase. *Energy Environ. Sci.* **12**, 1938-1949 (2019).
11. Li W, Wang K, Zhou M, Zhan H, Cheng S, Jiang K. Advanced low-cost, high-voltage, long-life aqueous hybrid sodium/zinc batteries enabled by a dendrite-free zinc anode and concentrated electrolyte. *ACS Appl. Mater. Interfaces* **10**, 22059-22066 (2018).
12. Deng C, *et al.* A sieve-functional and uniform-porous kaolin layer toward stable zinc metal anode. *Adv. Funct. Mater.* **30**, 2000599 (2020).
13. Zhao K, *et al.* Ultrathin surface coating enables stabilized zinc metal anode. *Adv. Mater. Interfaces* **5**, 1800848 (2018).
14. Cao L, Li D, Deng T, Li Q, Wang C. Hydrophobic organic electrolyte protected Zn anodes for aqueous Zn batteries. *Angew. Chem. Int. Ed.* **59**, 19292-19296 (2020).
15. Kang LT, *et al.* Nanoporous CaCO₃ coatings enabled uniform Zn stripping/plating for long-life zinc rechargeable aqueous batteries. *Adv. Energy Mater.* **8**, 1801090 (2018).

16. Zhang Q, *et al.* Revealing the role of crystal orientation of protective layers for stable zinc anode. *Nat. Commun.* **11**, 3961 (2020).
17. Pan H, *et al.* Reversible aqueous zinc/manganese oxide energy storage from conversion reactions. *Nat. Energy* **1**, 16039 (2016).
18. Wang J, Wang J-G, Liu H, Wei C, Kang F. Zinc ion stabilized MnO₂ nanospheres for high capacity and long lifespan aqueous zinc-ion batteries. *J. Mater. Chem. A* **7**, 13727-13735 (2019).
19. Han S-D, *et al.* Mechanism of Zn insertion into nanostructured δ -MnO₂: a nonaqueous rechargeable Zn metal battery. *Chem. Mater.* **29**, 4874-4884 (2017).
20. Bhojate S, Mhin S, Jeon JE, Park K, Kim J, Choi W. Stable and high-energy-density Zn-ion rechargeable batteries based on a MoS₂-coated Zn anode. *ACS Appl. Mater. Interfaces* **12**, 27249-27257 (2020).
21. Hao J, *et al.* An in-depth study of Zn metal surface chemistry for advanced aqueous Zn-ion batteries. *Adv. Mater.* **32**, e2003021 (2020).
22. Dong L, *et al.* Extremely safe, high-rate and ultralong-life zinc-ion hybrid supercapacitors. *Energy Storage Mater.* **13**, 96-102 (2018).
23. Wang Z, *et al.* A metal-organic framework host for highly reversible dendrite-free zinc metal anodes. *Joule* **3**, 1289-1300 (2019).
24. Cui Y, *et al.* Quasi-solid single Zn-ion conductor with high conductivity enabling dendrite-free Zn metal anode. *Energy Storage Mater.* **27**, 1-8 (2020).
25. Zou K, *et al.* Highly stable zinc metal anode enabled by oxygen functional groups for advanced Zn-ion supercapacitors. *Chem. Commun.* **57**, 528-531 (2021).
26. Safi I. Recent aspects concerning DC reactive magnetron sputtering of thin films: A review. *Surf. Coat. Technol.* **127**, 203-219 (2000).
27. Du C-F, Liang Q, Luo Y, Zheng Y, Yan Q. Recent advances in printable secondary batteries. *J. Mater. Chem. A* **5**, 22442-22458 (2017).
28. Liu Y, *et al.* Making Li-metal electrodes rechargeable by controlling the dendrite growth direction. *Nat. Energy* **2**, 1-10 (2017).

29. Cheng XB, Zhang R, Zhao CZ, Zhang Q. Toward safe lithium metal anode in rechargeable batteries: A Review. *Chem. Rev.* **117**, 10403-10473 (2017).
30. Liu H, Peng D, Xu T, Cai K, Sun K, Wang Z. Porous conductive interlayer for dendrite-free lithium metal battery. *J. Energy Chem.* **53**, 412-418 (2021).
31. Zou P, *et al.* A periodic “self-correction” scheme for synchronizing lithium plating/stripping at ultrahigh cycling capacity. *Adv. Funct. Mater.* **30**, 1910532 (2020).
32. Chen X, *et al.* A “dendrite-eating” separator for high-areal-capacity lithium-metal batteries. *Energy Storage Mater.* **31**, 181-186 (2020).
33. Foroozan T, Yurkiv V, Sharifi-Asl S, Rojaee R, Mashayek F, Shahbazian-Yassar R. Non-dendritic Zn electrodeposition enabled by zincophilic graphene substrates. *ACS Appl. Mater. Interfaces* **11**, 44077-44089 (2019).
34. Cui M, *et al.* Quasi-isolated Au particles as heterogeneous seeds to guide uniform zn deposition for aqueous zinc-ion batteries. *ACS Appl. Energy Mater.* **2**, 6490-6496 (2019).
35. Liu Y, Ji C, Su X, Xu J, He X. Electromagnetic and microwave absorption properties of Ti_3SiC_2 powders decorated with Ag particles. *J. Alloys Compd.* **820**, 153154 (2020).

References

Reviewer #2

Comments to the Author

Dendrites growth is a long-standing issue for the plating/stripping process of metal anodes, such as alkali metal and zinc anodes, which significantly impedes their practical application. This manuscript demonstrates a novel Sn-modified separator for zinc/sodium metal anodes to realize the long-term cycling at simultaneous high current density and charge/discharge capacity, outperforming the performances of previous works. The Sn element is chosen due to its decent stability and good zincophilicity originated from the electrochemical alloy reaction between Sn and Zn. Theoretical calculations and experimental results indicate that the improved plating/stripping behavior comes from the even electric field distribution and face-to-face Zn deposition. This research presents a thoroughly detailed study, providing an exciting approach to suppress metal dendrites growth. Thus, I recommend this manuscript to be published after addressing a few issues listed as follows.

We appreciate these positive comments from the reviewer.

1. The authors prepared the metal modified Ti foils that are used to test the nucleation overpotential of various metal elements by mixing metal powders, Super P, and PVDF (8:1:1 of weight ratio). The nucleation overpotential of pure Super P modified Ti foils should be supplied to serve as the control group.

Response: As suggested, Super P modified Ti foil (denoted as SP-Ti) is fabricated by casting the Super P slurry onto the Ti foil using the doctor blade method. The slurry is prepared by uniformly mixing 90 wt% Super P and 10 wt% polyvinylidene difluoride (PVDF) in N-methyl-2-pyrrolidone solvent. After drying at 60 °C for 12 h, SP-Ti foils are punched into circular disks (diameter of 12 mm) prior to use. The nucleation overpotential (η) of SP-Ti current collector is evaluated, as seen

in **updated Figure S4**. η of SP-Ti current collector is 38 mV, which is comparable to the 44 mV of the bare Ti current collector. This indicates that the influence of Super P on the nucleation overpotential is minor. The corresponding discussions have been added to the revised manuscript (**Page 19**) and supporting information (**Page S6**).

(Updated) Figure S4 The nucleation overpotential of Ti, metal-Ti and SP-Ti current collectors.

2. *The focus of this manuscript is to realize the stable cycling at high current density and cycling capacity. However, there are insufficient discussions to explain the disadvantages and failure mechanism at such a rigorous test condition. Authors should provide more discussions in the introduction and Figure 5 on this aspect.*

Response: Thanks for the reviewer's valuable suggestion. It is well known that during Zn deposition, an uneven electric field distribution inevitably forms at the pores of the separator (**updated Figures 1a, c**)^{1,2}. In addition, both Zn^{2+} and the electric field tend to concentrate at the protuberances with high surface energy³. Therefore, during the deposition, zinc nucleation and growth prefer to occur at such tips (i.e., "tip effect")³, resulting in inhomogeneous Zn deposition. The formed Zn protrusions further increase the local electric field intensity around them, leading to the evolution of Zn protuberances into Zn dendrites upon cycling. The inhomogeneous deposition of Zn is exacerbated at higher current densities, as the Zn^{2+} around the electrode/electrolyte interface are rapidly depleted^{4,5,6}. Subsequently, the dendrites are formed and quickly render a short circuit of cells. Higher cycling capacity would bring about greater volume change, which leads to the pulverization of Zn foil and may make Zn dendrites lose contact with

the electrodes, becoming “dead Zn”^{3,7}. This would finally give rise to a low reversibility for the Zn deposition/stripping process.

As shown in **updated Figure 5**, rapid cell failure is observed in the cell using pristine separator at 5 mA/cm² for 5 mAh/cm² and 10 mA/cm² for 10 mAh/cm². This is due to the rampant dendrites growth induced by the local electric field intensity and depleted Zn²⁺ concentration at the electrode/electrolyte interface. The SEM images after cycling provide concrete evidence (**original Figure S10**). A rugged surface with many protrusions is observed for the Zn anode using pristine separator after 1 cycle due to the inhomogeneous Zn deposition. Moreover, the cycled Zn evolves into a looser and rougher structure after 20 cycles. By contrast, cells with Sn-coated separator presents a significant improved stability even at extremely high current density and cycling capacity. For example, an exceptional Zn plating/stripping life of 500 h could be achieved at 10 mA/cm² and 10 mAh/cm², indicating its excellent potential for practical application. The reason for such exciting enhancements is due to the synergistic effects of uniform electric field (retard the growth of Zn dendrites) and face-to-face growth (eliminate the inevitably formed Zn dendrites). The dendrite-free morphologies are also demonstrated by SEM images. Dense Zn metal is observed on Sn-coated separator after 1 cycle and 20 cycles (**original Figure S10**). The Zn depositions growing from anode and separator are supposed to meet and merge upon cycling, resulting in a compact Zn metal layer with a change in the Zn growth direction. This is confirmed by the dendrite-free morphologies on both the anode and Sn-coated separator surface even after 200 cycles (**original Figure S11**). The merging of Zn grown from the anode and the separator will change the further growth direction of the Zn from perpendicular to parallel to the separator and greatly eliminate the hazard of short circuit induced by the dendrite growth. The corresponding discussions have been added to the revised manuscript (**Page 2 and 9**).

(Updated Figure 1) Theoretical calculation and protection mechanism of modified separator.

(a) Electric field distribution with the pristine separator. (b) Electric field distribution in the non-contact region of the modified separator and the anode. (c) Schematic illustration of Zn deposition with the pristine separator. (d) Schematic illustration of Zn deposition in the contact region of the modified separator and the anode.

(Updated) Figure 5 The electrochemical performance of Zn metal batteries. The cycling performance of Zn/Zn cells using pristine separator and Sn-coated separator tested at **(a)** 2 mA/cm² and 2 mAh/cm², **(b)** 5 mA/cm² and 5 mAh/cm² and **(c)** 10 mA/cm² and 10 mAh/cm². **(d)** Comparison of cycling performance (cumulative capacity versus per-cycle areal capacity) in this work and previously reported works. The electrochemical performance of Zn||MnO₂ batteries using pristine separator and the Sn-coated separator: **(e)** CV at a scan rate of 0.1 mV/s (second cycle); **(f)** Cycling performance at 0.3 A/g, with Zn foil as anode.

(Original) Figure S10 SEM images of the electrodes after Zn deposition at 5 mA/cm² and 5 mAh/cm². Cycled Zn using pristine separator after **(a)** 1 cycle and **(b)** 20 cycles. Zn using Sn-coated separator after **(c)** 1 cycle and **(d)** 20 cycles. Zn on cycled Sn-coated separator after **(e)** 1 cycle and **(f)** 20 cycles.

(Original) Figure S11 SEM images of **(a)** cycled Zn using Sn-coated separator and **(b)** cycled Zn on Sn-coated separator after 200 cycles at 5 mA/cm² and 5 mAh/cm².

3. In Figures 4a and b, the morphologies of Zn deposition using a pristine separator are nonuniform. What is the reason behind this phenomenon?

Response: Under a deposition capacity of 1 mAh/cm^2 , loose structures with inhomogeneous Zn chaotic clusters are observed on the Ti current collector using a pristine separator (**updated Figure 4a**). With an increased deposition capacity of 4 mAh/cm^2 , Zn maintains similar dreadful morphologies with sharp tips presented (**updated Figure 4b**), which might pierce the separator upon cycling. The reasons are as follows: It is well known that during Zn deposition, an uneven electric field distribution inevitably forms at the pores of the separator (**updated Figures 1a, c**)^{1,2}. In addition, both Zn^{2+} and the electric field tend to concentrate at the protuberances with high surface energy³. Therefore, during the deposition, zinc nucleation and growth prefer to occur at such tips (i.e., “tip effect”)³, resulting in inhomogeneous Zn deposition. The formed Zn protrusions further increase the local electric field intensity around them, leading to the evolution of Zn protuberances into Zn dendrites upon cycling. The corresponding discussions have been added to the revised manuscript (**Page 9**).

(New) Figure 4 SEM images of the electrodes after Zn deposition at 1 mA/cm^2 . Zn deposition on Ti current collector using pristine separator with a cycling capacity of (a) 1 mAh/cm^2 and (b) 4

mAh/cm². Zn deposition on Ti current collector using Sn-coated separator with a cycling capacity of (c) 1 mAh/cm² and (d) 4 mAh/cm². Zn deposition on Sn-coated separator with a cycling capacity of (e) 1 mAh/cm² and (f) 4 mAh/cm².

4. Why the overpotential decreases with the cycles in Figure 5, especially for Sn coated separator?

Response: The decrease in overpotential with cycling is attributed to the fact that the passivation layer on the pristine Zn foil is destroyed upon cycling⁸. For Zn/Zn cells with Sn-coated separator, Zn metal is concurrently deposited on both anode and separator. Zn metal on Sn-coated separator may help to build more continuous electrical conductivity layer compared with pristine Sn-coated separator and achieve a better electrical conductivity, in turn facilitating the Zn deposition process and decreasing the overpotential. The corresponding discussions have been added to the revised supporting information (**Page S15**).

5. In Figure 5f, why the CEs of full cells using Sn coated separator outperform those of pristine separator? Moreover, please explain the reasons for the decreased CE of the cells using a pristine separator as the discharge capacity drops.

Response: The higher CEs of full cells using the Sn-coated separator are ascribed to the better reversibility of Zn metal anode, as confirmed by the CEs of Ti/Zn cells. As shown in **new Figure S11**, the CE values of cells with Sn-coated separator are higher and more stable than that using the pristine separator at both testing conditions. Moreover, a rapid short circuit is observed for the cell with pristine separator, indicating a rampant growth of Zn dendrites during the deposition/stripping process. Therefore, the Sn-coated separator enables a better Zn reversibility and restrains Zn dendrites growth. The CE decreases when the discharge capacity drops in the cell using pristine separator, which demonstrates that the inferior reversible Zn deposition/stripping behavior accounts for the rapid capacity fade of full cells. The corresponding discussions have been added to the revised manuscript (**Page 10 and 11**) and supporting information (**Page S13 and S14**).

(New) Figure S11 CEs of Ti/Cu cells and detailed deposition/stripping voltage curves using pristine separator and Sn-coated separator at **(a)** 5 mA/cm² and 5 mAh/cm² and **(b)** 10 mA/cm² and 10 mAh/cm².

References

1. Zhao CZ, *et al.* An ion redistributor for dendrite-free lithium metal anodes. *Sci. Adv.* **4**, eaat3446 (2018).
2. Qin Y, *et al.* Advanced filter membrane separator for aqueous zinc-ion batteries. *Small* **16**, e2003106 (2020).
3. Yang Q, *et al.* Dendrites in Zn-based batteries. *Adv. Mater.* **32**, e2001854 (2020).
4. Cheng XB, Zhang R, Zhao CZ, Zhang Q. Toward safe lithium metal anode in rechargeable batteries: A Review. *Chem. Rev.* **117**, 10403-10473 (2017).

5. Yi Z, Chen G, Hou F, Wang L, Liang J. Strategies for the stabilization of Zn metal anodes for Zn-ion batteries. *Adv. Energy Mater.* **11**, 2003065 (2020).
6. Xie C, Li Y, Wang Q, Sun D, Tang Y, Wang H. Issues and solutions toward zinc anode in aqueous zinc-ion batteries: A mini review. *Carbon Energy* **2**, 540-560 (2020).
7. Lin D, Liu Y, Cui Y. Reviving the lithium metal anode for high-energy batteries. *Nat. Nanotechnol.* **12**, 194-206 (2017).
8. Kang LT, *et al.* Nanoporous CaCO₃ coatings enabled uniform Zn stripping/plating for long-life zinc rechargeable aqueous batteries. *Adv. Energy Mater.* **8**, 1801090 (2018).

Reviewer #3

Comments to the Author

This manuscript reports a concept of preventing dendrite formation in Zn and Na batteries by introducing an Sn-coated separator, leading to a uniform plating of Zn (or Na) between the separator and metal anode, thus enabling high capacity and high-rate performance. The manuscript is easy to read and delivers the concept and results clearly. It will give inspiring ideas to researchers in these areas to develop Zn or Na batteries further. However, there are a few issues to be cleared before it is considered for publication. My comments are as follows.

We appreciate much the positive comments from the reviewer.

1. Strictly speaking, though the concept the authors wanted to deliver is obvious, the figure in the middle of Figure 1d is not right in the sense that Zn cannot be plated on the points without an electronic connection to the current collector. The coated Sn particles must directly contact the Zn anode or indirectly through the coated Sn network on the separator's surface. The middle figure shows no contact with the Zn anode, and the particles on the separator seem to be separated from each other.

Response: We fully agree with the comment. Indeed, there must be direct contact between the Sn layer and anode for electron transfer. Therefore, we re-draw **Figure 1** to demonstrate the mechanism more clearly and accurately. As shown in **updated Figure 1**, the contact between Zn anode and Sn-coated separator and the connection between particles are illustrated. These modifications have been added to the revised manuscript (**Page 5**).

(Updated Figure 1) Theoretical calculation and protection mechanism of modified separator.

(a) Electric field distribution with the pristine separator. **(b)** Electric field distribution in the non-contact region of the modified separator and the anode. **(c)** Schematic illustration of Zn deposition with the pristine separator. **(d)** Schematic illustration of Zn deposition in the contact region of the modified separator and the anode.

2. Fig 2a and Fig 3d show that the stable Zn-Sn alloying (or Zn plating) potential is around -0.02 V. In contrast, Fig 2b shows the reduction potential is just below 0.3 V, which might be explained as a Zn-Sn alloy-formation reaction. The question is why the two voltages are so different.

Response: We apologize for not explaining it clearly in the previous version. As shown in **original Figure 2a** and **original Figure 3d**, the plating potential for Zn^{2+} is around -0.02 V. This value represents an overpotential required for overcoming the barrier to reduce Zn^{2+} into Zn metal, and the involved reaction is $Zn^{2+} + 2e \rightarrow Zn$. While the reduction peak in **original Figure 2b** refers to the electrochemical alloying process between Sn and Zn. A value of just below 0.3 V is required to reduce Zn^{2+} and form an alloy with $Sn^{1,2}$, and the involved reaction is: $xZn^{2+} + Sn + 2xe^- \rightarrow SnZn_x$. The strong affinity of Sn to Zn greatly decreases the energy barrier required for the subsequent reduction of Zn^{2+} towards Zn. The corresponding discussions have been added to the revised supporting information (**Page S6 and S7**).

(Original) Figure 2 The alloy reaction between Sn and Zn. (a) The nucleation overpotential of Zn on Ti and Sn-Ti current collectors. (b) CV of Sn-Ti/Zn cell at a scan rate of 0.2 mV/s. (c) The galvanostatic discharge curve of Sn-Ti/Zn cell at 0.08 mA/cm². (d) XPS spectra of Sn 3d and Zn 2p for pristine and discharged Sn.

(Original) Figure 3 Characterization of Sn-coated separator. The SEM images of (a) pristine separator and (b) Sn-coated separator with the corresponding optical photos in the insets. (c) The

XRD spectrum of Sn-coated separator. (d) The nucleation overpotential of Ti/Zn cell with pristine separator and Sn-coated separator.

3. Fig 2c shows the capacity of 21 mAh/g. Based on the gram of what material was it calculated? I suppose it is Sn, which should be explicitly noted in the text.

Response: Thanks for pointing this out. The capacity of 21 mAh/g is calculated based on the gram of Sn. The modification has been added to the revised manuscript (**Page 6**).

4. Fig 5b. Even for the Sn-coated separator case, there is fluctuation during cycling? Does it imply a micro short-circuit?

Response: To clarify this issue, we have re-tested Zn/Zn cells using Sn-coated separator under two testing conditions (5 mA/cm^2 for 5 mAh/cm^2 and at 10 mA/cm^2 for 10 mAh/cm^2) at constant temperature. As shown in **updated Figures 5b, c** and **new Figure S13**, cell with Sn-coated separator delivers stable Zn deposition/stripping curves upon long-term cycling at both 5 mA/cm^2 for 5 mAh/cm^2 and at 10 mA/cm^2 for 10 mAh/cm^2 . Still, a minor fluctuation in the voltage profiles is observed. We speculate the fluctuation is due to the concurrent deposition of Zn on both anode and Sn-coated separator. The overpotentials required for depositing Zn^{2+} on the anode and the Sn-coated separator are different. The deposition amount of Zn on anode and Sn-separator slightly varies with cycles, which may result in the various voltage potentials for different cycles. The corresponding discussions have been added to the revised manuscript (**Page 11**) and supporting information (**Page S15 and S16**).

(Updated) Figure 5 The electrochemical performance of Zn metal batteries. The cycling performance of Zn/Zn cells using pristine separator and Sn-coated separator tested at **(b)** 5 mA/cm² and 5 mAh/cm² and **(c)** 10 mA/cm² and 10 mAh/cm².

(New) Figure S13 Reproducible experimental data for the cycling performances of Zn/Zn cells using pristine separator and Sn-coated separator tested at **(a)** 5 mA/cm² and 5 mAh/cm² and **(b)** 10 mA/cm² and 10 mAh/cm².

5. Fig 5f. Why the capacity for the Sn-coated separator 10:1 case (for instance) gradually increased up to 60 cycles?

Response: This phenomenon arises from the instability of prepared MnO₂ cathode rather than the anode, which is evidenced by the obvious capacity increase in the first 10 cycles for the full cells with Zn foil as the anode (**original Figure S13**). The previously used MnO₂ cathode materials are fabricated using the hydrothermal method³.

To obtain stable and reliable MnO₂ cathode materials, we turn to employ other method (chemical reaction according to the previous report⁴) and try our utmost effort to synthesize them. Concretely, 2.5 mL of 1.0 M H₂SO₄ and 158 mg of KMnO₄ were mixed with deionized water (30 mL) under magnetic stirring until dissolving. Subsequently, 95 mg of zinc powders were added into the solution and were magnetically stirred at 60 °C for 6 h. The production was collected by filtration and washed with deionized water until pH was higher than 6. Finally, it was dried at 80 °C for 12 h. The XRD and SEM were respectively used to observe the phase structure and microstructure morphologies of as-received sample. As shown in **new Figure S25a**, the diffraction peaks are consistent with the birnessite-type MnO₂ with a layered structure (JCPDS#43-1456)^{4,5}. SEM images show that the morphologies of such MnO₂ is flower-like nanospheres (**new Figures S25b, c**), which agrees with the literature⁴.

Subsequently, the performance of full cells is re-collected using this MnO₂ cathode. It is found that the phenomenon of dramatical increased capacity in initial cycles is greatly alleviated. The CV curves of full cells using pristine separator and Sn-coated separator are compared in **updated Figure 5e**. They present the same Mn-ion redox peaks, which agrees well with the previous works⁴. The full cell with Sn-coated separator shows lower oxidation potential, higher reduction potential and peak current than that with pristine separator, suggesting improved reaction kinetics for Sn-coated separator^{6,7,8}. Supportive evidence could be found at the rate performances of full cells **updated Figure S26**. At a high current density of 0.75 A/g (equivalent to 2 C), a full cell with the Sn-coated separator can provide a discharge capacity equivalent to approximately 200% of that with the pristine separator (107 mAh/g vs. 53 mAh/g).

As shown in **updated Figure 5f**, full cell with the Sn-coated separator presents a stable cycle life with a discharge capacity of ~200 mAh/g after 600 cycles. Turning to the pristine separator, the discharge capacity is ~159 mAh/g after 130 cycles and then gradually drops to ~94 mAh/g after 260 cycles. More critically, using a pre-set amount of Zn as the anode, the cycling performance of the full cell is further evaluated at the specific negative-to-positive electrode capacity (N:P) ratios of 10:1. The discharge capacity of the full cell with the pristine separator deteriorates rapidly after about 60 cycles, with a capacity of merely ~63 mAh/g after 80 cycles (**updated Figure S27**), demonstrating the poor Zn reversibility. On the contrary, the highly improved cycle stability is achieved over 180 cycles (discharge capacity of ~145 mAh/g) on the full cell with the Sn-coated separator.

Therefore, the improved stability is realized on full cells using Sn-coated separator after re-preparing the MnO₂ cathode. In the future, we will further optimize the synthesis technique of MnO₂ cathodes and fabricate other cathode materials to match this highly stable Zn metal anode achieved by the Sn-coated separator. The corresponding discussions have been added to the revised manuscript (**Page 14, 15 and 19**) and supporting information (**Page S28 and S29**).

(Original) Figure S13 The cycle performances of Zn||MnO₂ batteries using pristine separator and Sn-coated separator at 0.3 A/g, with Zn foil as anode.

(New) Figure S25 (a) The XRD spectrum and (b, c) SEM images of newly prepared MnO₂ materials.

(Updated) Figure 5 The electrochemical performances of Zn||MnO₂ batteries using pristine separator and Sn-coated separator: (e) CV at a scan rate of 0.1 mV/s (second cycle); (f) Cycling performances at 0.3 A/g, with Zn foil as anode.

(Updated) Figure S26 The rate performances of Zn||MnO₂ batteries using pristine separator and Sn-coated separator.

(Updated) Figure S27 The cycling performances of Zn||MnO₂ batteries using pristine separator and Sn-coated separator at 0.3 A/g with N:P ratios of 10:1.

6. Typos:

Page 4: to guild -- > build (?)

Page 5: voltage dip  tip

Response: We are sorry for the typos and have corrected them in the revised manuscript. The modifications have been added on the revised manuscript (**Page 4 and 6**).

References

1. Yan K, *et al.* Selective deposition and stable encapsulation of lithium through heterogeneous seeded growth. *Nat. Energy* **1**, 16010 (2016).
2. Tang S, *et al.* Stable Na plating and stripping electrochemistry promoted by in situ construction of an alloy-based sodiophilic interphase. *Adv. Mater.* **31**, e1807495 (2019).
3. Pan H, *et al.* Reversible aqueous zinc/manganese oxide energy storage from conversion reactions. *Nat. Energy* **1**, 16039 (2016).
4. Wang J, Wang J-G, Liu H, Wei C, Kang F. Zinc ion stabilized MnO₂ nanospheres for high capacity and long lifespan aqueous zinc-ion batteries. *J. Mater. Chem. A* **7**, 13727-13735 (2019).
5. Han S-D, *et al.* Mechanism of Zn insertion into nanostructured δ -MnO₂: a nonaqueous rechargeable Zn metal battery. *Chem. Mater.* **29**, 4874-4884 (2017).
6. Bhoyate S, Mhin S, Jeon JE, Park K, Kim J, Choi W. Stable and high-energy-density Zn-ion rechargeable batteries based on a MoS₂-coated Zn anode. *ACS Appl. Mater. Interfaces* **12**, 27249-27257 (2020).
7. Deng C, *et al.* A sieve-functional and uniform-porous kaolin layer toward stable zinc metal anode. *Adv. Funct. Mater.* **30**, 2000599 (2020).
8. Hao J, *et al.* An in-depth study of Zn metal surface chemistry for advanced aqueous Zn-ion batteries. *Adv. Mater.* **32**, e2003021 (2020).

Reviewer #1 (Remarks to the Author):

I'm OK with the revised version.

Reviewer #2 (Remarks to the Author):

The authors have tried their best to address all my raised issues and almost the issues raised by other reviewers. The quality of this manuscript has been greatly improved and it can be accepted by this Journal in the present form.

Reviewer #3 (Remarks to the Author):

Most of the reviewers' comments are well addressed in the revised manuscript, which is considerably improved. I support its publication in this journal.

Author response to Reviewers' Comments:

Reviewer #1 (Remarks to the Author):

I'm OK with the revised version.

Reply: We appreciate the reviewer's positive comment.

Reviewer #2 (Remarks to the Author):

The authors have tried their best to address all my raised issues and almost the issues raised by other reviewers. The quality of this manuscript has been greatly improved and it can be accepted by this Journal in the present form.

Reply: We are grateful for the reviewer's recommendation.

Reviewer #3 (Remarks to the Author):

Most of the reviewers' comments are well addressed in the revised manuscript, which is considerably improved. I support its publication in this journal.

Reply: Thank you very much for your great encouragement to our work.